# Mice lacking the mitochondrial exonuclease MGME1 accumulate mtDNA deletions without developing progeria

Stanka Matic[1], Min Jiang[1], Thomas J. Nicholls[2], Jay P. Uhler[2], Caren Dirksen-Schwanenland[1], Paola Loguercio Polosa[3], Marie-Lune Simard[1], Xinping Li[4], Ilian Atanassov [4], Oliver Rackham [5], Aleksandra Filipovska[5], James B. Stewart [1], Maria Falkenberg[2], Nils-Göran Larsson [1,6] & Dusanka Milenkovic [1]

Replication of mammalian mitochondrial DNA (mtDNA) is an essential process that requires high fidelity and control at multiple levels to ensure proper mitochondrial function. Mutations in the mitochondrial genome maintenance exonuclease 1 (*MGME1*) gene were recently reported in mitochondrial disease patients. Here, to study disease pathophysiology, we generated *Mgme1* knockout mice and report that homozygous knockouts develop depletion and multiple deletions of mtDNA. The mtDNA replication stalling phenotypes vary dramatically in different tissues of *Mgme1* knockout mice. Mice with MGME1 deficiency accumulate a long linear subgenomic mtDNA species, similar to the one found in mtDNA mutator mice, but do not develop progeria. This finding resolves a long-standing debate by showing that point mutations of mtDNA are the main cause of progeria in mtDNA mutator mice. We also propose a role for MGME1 in the regulation of replication and transcription termination at the end of the control region of mtDNA.

[1] Department of Mitochondrial Biology, Max Planck Institute for Biology of Ageing, Cologne 50931, Germany. [2] Department of Medical Biochemistry and Cell Biology, University of Gothenburg, Gothenburg 405 30, Sweden. [3] Department of Biosciences, Biotechnologies and Biopharmaceutics, University of Bari Aldo Moro, Bari 70125, Italy. [4] Proteomics Core Facility, Max Planck Institute for Biology of Ageing, Cologne 50931, Germany. [5] Harry Perkins Institute of Medical Research Centre for Medical Research and School of Molecular Sciences, The University of Western Australia, Nedlands WA 6009, Australia. [6] Department of Medical Biochemistry and Biophysics, Karolinska Institutet, 171 77 Stockholm, Sweden. Correspondence and requests for materials should be addressed to N.-G.L. (email: nils-goran.larsson@ki.se) or to D.M. (email: dmilenkovic@age.mpg.de)

Mitochondrial diseases represent the most common group of inherited metabolic diseases in humans with a prevalence of about 1 in 5000[1]. These genetically heterogeneous disorders can be caused by mutations in mitochondrial DNA (mtDNA) or in nuclear genes that encode proteins with mitochondrial function[2]. Mutations in nuclear genes cause mtDNA instability resulting in mtDNA depletion or accumulation of deletions and/or point mutations, ultimately leading to impaired oxidative phosphorylation (OXPHOS). The vast majority of mutations causing human mtDNA instability map to genes encoding proteins involved in mtDNA replication, e.g., the catalytic subunit of mtDNA polymerase (*POLGA*)[3], the accessory subunit of mtDNA polymerase (*POLGB*)[4], the replicative helicase (*TWNK*)[5], DNA replication helicase/nuclease 2 (*DNA2*)[6], mitochondrial genome maintenance exonuclease 1 (*MGME1*)[7] and ribonuclease H1 (*RNASEH1*)[8], or nucleotide pool regulation, e.g., thymidine phosphorylase (*TP*)[9], mitochondrial thymidine kinase

(*TK2*)[10], deoxyguanosine kinase (*DGUOK*)[11], ATP-dependent succinate-CoA ligase (*SUCLA2*)[12], and GTP-dependent succinate-CoA ligase (*SUCLG1*)[1,13,14]. Mitochondrial disorders are highly variable in disease severity and clinical presentation, and display tissue-specific manifestations. The basis for the clinical heterogeneity and tissue specificity of mutations in these ubiquitously expressed genes is unknown. The fact that mtDNA maintenance defects primarily involve high energy demanding tissues, such as brain and muscle, can only partially explain the tissue specificity seen in mitochondrial disorders. Other organs are also frequently affected causing, e.g., cardiomyopathy, diabetes mellitus, liver dysfunction, and optic neuropathy[1]. Extensive in vitro work has led to significant progress in our understanding of the biochemical processes underlying mtDNA maintenance disorders[4,15,16], but animal models are nevertheless essential to understand the wide range of phenotypes and secondary metabolic consequences of mtDNA instability in different tissues[17].

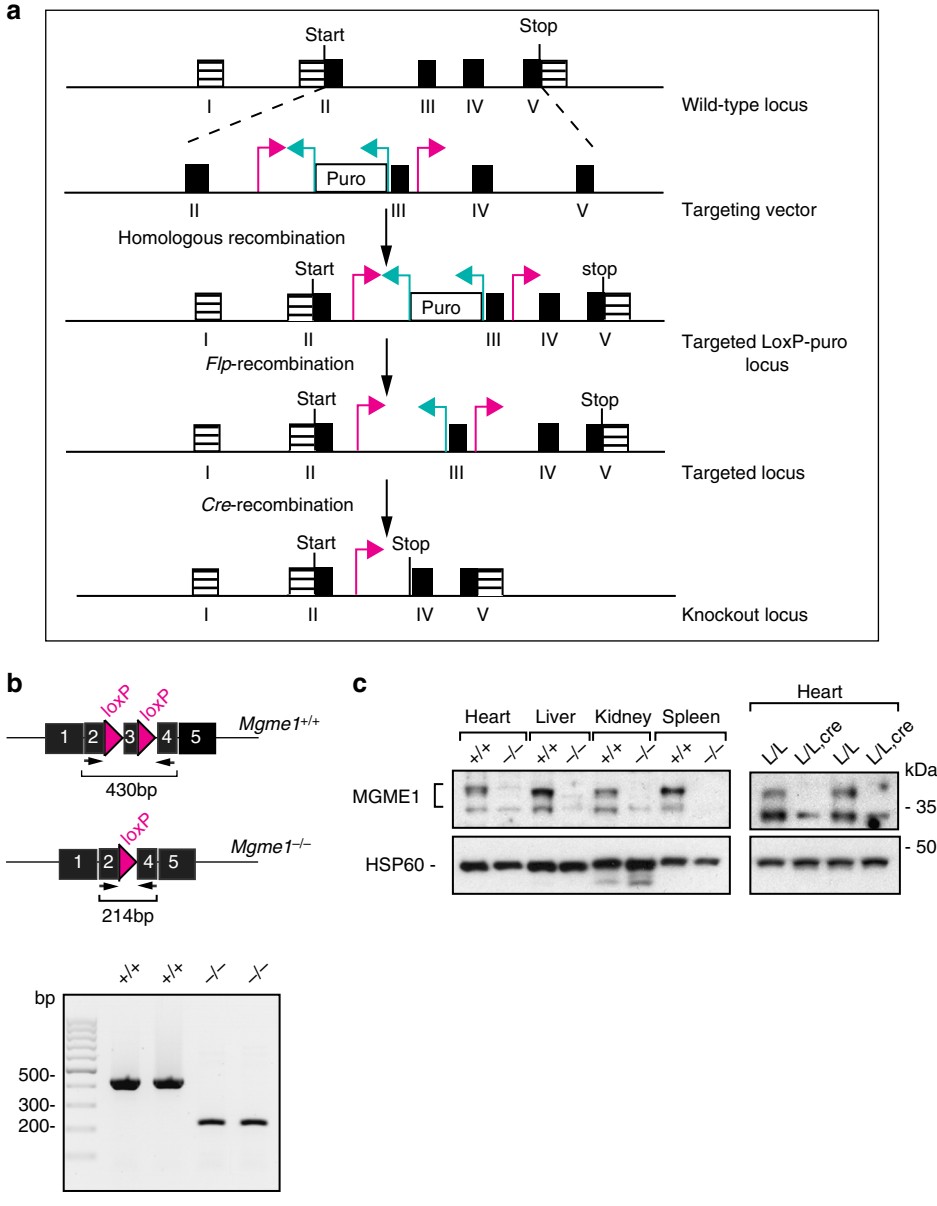

**Fig. 1** Conditional knockout of *Mgme1*. **a** Targeting strategy for disruption of the *Mgme1* gene. Magenta arrows loxP sequences; turquoise arrows *FRT* sites. **b** Schematic representation of *Mgme1* cDNA. RT-PCR analysis of *Mgme1* transcripts from control (+/+) and *Mgme1* knockout mice (−/−). **c** Western blot analysis of MGME1 levels in heart, liver, kidney, and spleen mitochondrial extracts of *Mgme1* knockout (−/−) and wild-type (+/+) mice and heart mitochondrial extracts of control (*L/L*) and tissue-specific knockout mice (*L/L, cre*). Mitochondrial HSP60 was used as a loading control

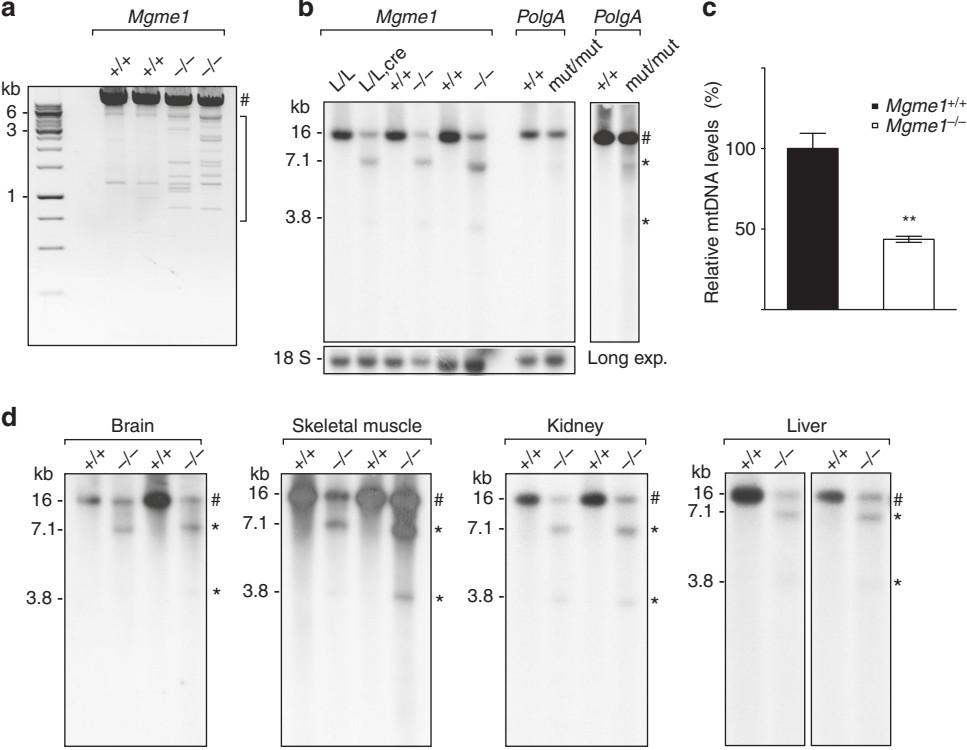

**Fig. 2** MtDNA depletion and accumulation of deletions in *Mgme1* homozygous and tissue-specific knockout mice in various tissues. **a** Long-extension PCR of heart-isolated total DNA from 8 weeks old controls (+/+) and *Mgme1* knockout mice (−/−) using primers amplifying major arc. # indicates product form full-length mtDNA, parentheses indicate deleted products. The bands are visualized by EtBr staining. **b** Southern blot analysis of SacI-digested heart mtDNA from control (*Mgme1^L/L^*) and tissue-specific knockout mice (*Mgme1^L/L^*, cre), wild-type (*Mgme1^+/+^*) and Mgme1 knockout mice (*Mgme1^−/−^*) and control (*PolgA^+/+^*) and mtDNA-mutator mice (*PolgA^mut/mut^*). Plasmid pAM1, containing whole mtDNA sequence was used as a probe. # full-length mtDNA; *deleted mtDNA molecules. **c** Southern blot quantification: ratio of full-length mtDNA signal to 18S loading control. Error bars represent the SEM. **P < 0,001, Statistical analysis were performed using Student's *t*-test. n = 3 biological replicates. **d** Southern blot analysis of SacI-digested mtDNA from brain, skeletal muscle, kidney, and liver tissue. (+/+), control; (−/−), Mgme1 knockout mice. Plasmid containing whole mtDNA sequence was used as a probe. # full-length mtDNA; *deleted mtDNA fragments

Various animal models have increased our understanding of progression and disease mechanisms in mtDNA maintenance disorders and this will eventually open new avenues for therapeutic intervention[3,10,18–21].

To gain further insight into diseases of defective mtDNA replication, we created a knockout mouse model for the recently described disease gene encoding MGME1 (also known as Ddk1)[7,22]. Loss-of-function mutations in *MGME1* were reported to cause a severe multisystem mitochondrial disorder in humans with depletion and rearrangements of mtDNA[7,23]. Patients with loss-of-function MGME1 mutations have progressive external ophthalmoplegia, skeletal muscle wasting/weakness, emaciation, respiratory distress, severe dilated cardiomyopathy, microcephalus, mental retardation, and severe gastrointestinal symptoms. Biochemically, MGME1 is a single-stranded DNA nuclease involved in processing of 5′ mtDNA ends generated during replication[7,22,24]. Loss of MGME1 expression, either in siRNA treated cells or in patient fibroblasts, leads to an accumulation of 7S DNA[7,23], which is the single-stranded DNA species formed by premature replication termination at the end of the control region of mtDNA[23], thus suggesting a role for MGME1 in repressing formation or increasing turnover of these molecules.

We have studied the in vivo mtDNA replication phenotypes associated with MGME1 deficiency in various mouse tissues of knockout mice. Although MGME1 is not essential for embryonic development, its loss leads to accumulation of multiple deletions and depletion of mtDNA in a range of different mouse tissues. Furthermore, our data show that MGME1 is involved in

regulation of heavy-strand replication and transcription termination. Remarkably, we report that MGME1 knockout mice display tissue-specific replication stalling patterns, with different tissues accumulating distinct replication intermediates, showing that this mouse model is a valuable tool to investigate tissue-specific pathology caused by the absence of MGME1.

## Results

**Generation of *Mgme1* knockout mice.** To study the in vivo function of the *Mgme1* gene, we generated a conditional knockout allele (Fig. 1a). The mutated locus (*Mgme1^+/loxP-pur^*) was transmitted through the mouse germline and the puromycin selection cassette was removed by mating with transgenic mice ubiquitously expressing the *Flp*-recombinase. Mice heterozygous for the loxP-flanked *Mgme1* allele (*Mgme1^+/loxP^*) were mated to mice ubiquitously expressing *cre*-recombinase (*β-actin-cre*) to obtain heterozygous *Mgme1* knockout (*Mgme1^+/−^*) mice. An intercross of heterozygous *Mgme1^+/−^* animals gave viable homozygous knockout *Mgme1^−/−^* pups at the approximate expected Mendelian ratios (genotyped pups n = 204; *Mgme1^−/−^* n = 36; *Mgme1^+/−^* n = 119; *Mgme1^+/+^*; n = 49). *Mgme1^−/−^* mice had a normal gross appearance and were followed until the age of 70 weeks (Supplementary Fig. 1). Reverse transcription (RT)-PCR analyses of the *Mgme1* mRNA confirmed the absence of sequences corresponding to exon 3 (Fig. 1b) and the MGME1 protein was absent on western blots (Fig. 1c) in all investigated tissues of *Mgme1^−/−^* mice. We also generated heart- and skeletal-muscle-

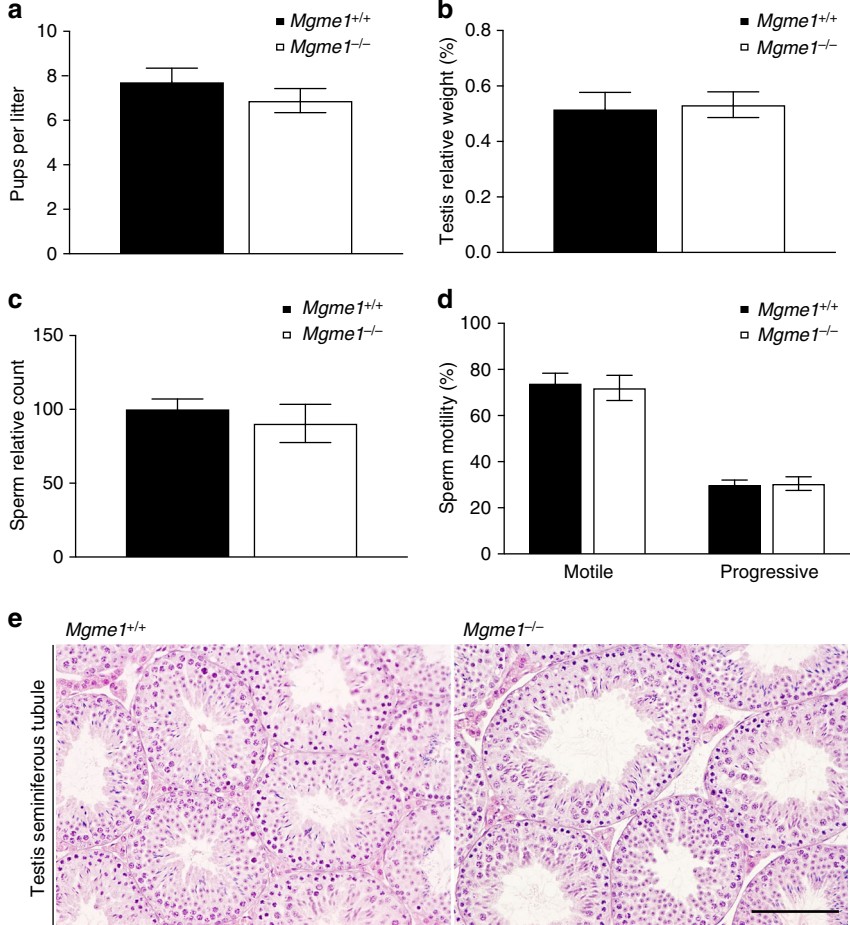

**Fig. 3** $Mgme1^{-/-}$ male mice are fertile and show normal testes morphology. **a** Pups per litter (data from 7 different males). **b** Relative testis weight. **c** Relative sperm count. **d** Sperm motility. **e** Testis seminiferous tubule structure. Scale bar, 100 μm. Data represent mean ± SEM. Statistical analyses were performed using Student's $t$-test. $n = 3$ biological replicates

specific $Mgme1$ knockout mice ($Mgme1^{loxP/loxP}$; $+/Ckmm$-cre) that had no obvious phenotype when followed until 12 months of age. The MGME1 protein was absent in heart of the $Mgme1^{loxP/loxP}$; $+/Ckmm$-cre animals (Fig. 1c). These results show that mice with both germline and tissue-specific knockout of $Mgme1$ are viable and appear healthy.

**Widespread distribution of deleted mtDNA in $Mgme1^{-/-}$ mice**. We performed long-extension PCR on total heart DNA to assess mtDNA integrity in $Mgme1^{-/-}$ mice (Fig. 2a). When we used a primer pair (Supplementary Fig. 2a) designed to amplify the major arc of mtDNA, which is the region most commonly affected by deletions, we found multiple shorter molecules on long-extension PCR, consistent with the presence of mtDNA rearrangements in $Mgme1^{-/-}$ mice (Fig. 2a). Moreover, Southern blot analyses of heart DNA from $Mgme1^{loxP/loxP}$; $+/Ckmm$-cre mice and various tissues of $Mgme1^{-/-}$ mice showed both mtDNA depletion and a prominent deletion of mtDNA (Fig. 2b–d). With restriction enzyme mapping, we established that the deleted species was a linear mtDNA molecule of ~11 kb that lacks sequences corresponding to the minor arc of mtDNA, which is the shorter region extending between the origin of replication of the leading (heavy, $O_H$) and lagging (light, $O_L$) strand of mtDNA. Consistent with the location of the deletion, both SacI (Fig. 2b, d and Supplementary Fig. 2d) and XhoI (Supplementary Fig. 2b) digested the linear mtDNA species into two fragments. In contrast, EagI, which only digests mtDNA in

the minor arc region, did not cut the linear mtDNA molecule (Supplementary Fig. 2c). Similar to the findings in $Mgme1^{-/-}$ mice, patients with pathogenic $MGME1$ mutations display multiple mtDNA deletions in skeletal muscle, blood and urine on long-extension PCR analyses[7], as well as mtDNA depletion and high levels of a linear subgenomic mtDNA molecule on Southern blot analyses of fibroblast mtDNA[7,23]. We have previously reported that the prematurely ageing mtDNA mutator ($PolgA^{mut/mut}$) mouse has high levels of a linear subgenomic mtDNA molecule in addition to high levels of point mutations in mtDNA[25,26]. The linear subgenomic mtDNA molecules are very similar in size (Fig. 2b, Supplementary Fig. 2b, d) in $Mgme1^{-/-}$ and $PolgA^{mut/mut}$ mice. However, $Mgme1^{-/-}$ mice accumulate more than double the amount of the 11 kb subgenomic fragment in comparison with $PolgA^{mut/mut}$ mice (Supplementary Fig. 2d, e). The finding of linear subgenomic mtDNA molecules of a similar size and extension in patients and mice lacking functional MGME1 and in mtDNA mutator mice expressing mutant POLGA suggests that both enzymes function in the same pathway and that there is a common mechanism for the formation of those subgenomic fragments[27].

**$Mgme1$ knockout mice do not age prematurely**. We assessed the mtDNA point mutation load in $Mgme1^{-/-}$ mice by using a high-fidelity polymerase to amplify mtDNA fragments followed by cloning and sequencing of individual clones. We detected wild-type levels of point mutations in liver of young (11 weeks of age)

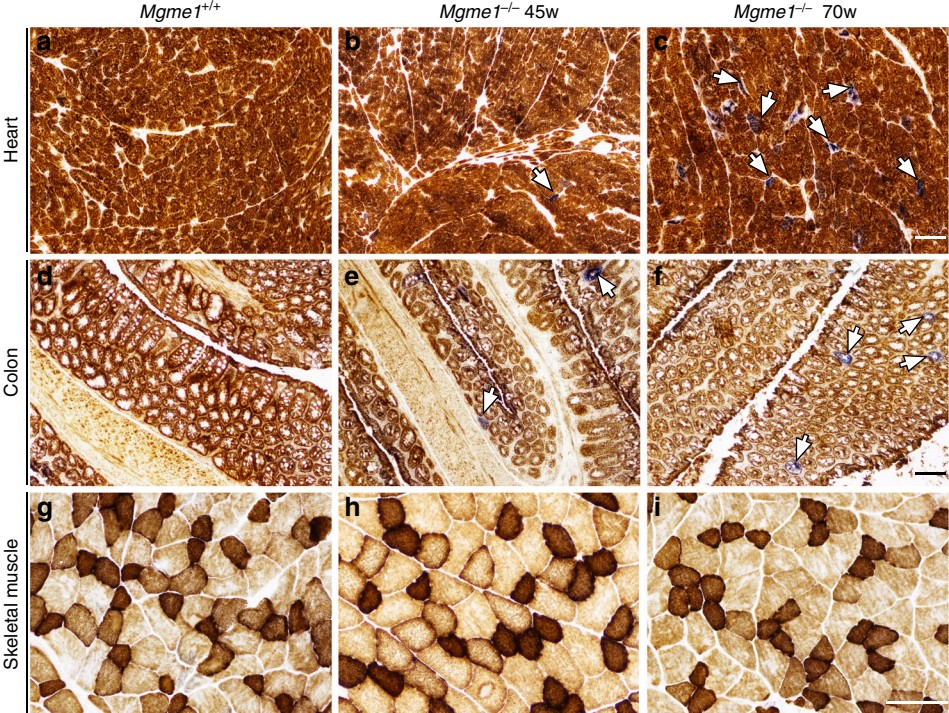

**Fig. 4** *Mgme1*$^{-/-}$ mice progressively accumulate COX negative cells in heart and colon tissue. COX/SDH staining of heart (**a–c**); colon (**d–f**); and skeletal muscle (**g–i**) from 45 and 70-week-old mice. White arrows indicate COX-defective cells. Scale bar, 100 μm. Three biological replicates per genotype

and old *Mgme1*$^{-/-}$ mice (70 weeks of age) (Supplementary Fig. 3a). Furthermore, we could not detect any significant changes in the load of mtDNA point mutations in spleen and skeletal muscle of 70 weeks old *Mgme1*$^{-/-}$ mice (Supplementary Fig. 3b). The increase in the point mutation load in mtDNA mutator mice is associated with premature onset of ageing-related phenotypes, such as decreased life span, male infertility and anemia[25,28,29]. In contrast, *Mgme1*$^{-/-}$ male mice are fertile with normal testis weight, normal sperm count, normal sperm motility and unaltered testis morphology (Fig. 3a–e).

At 20 weeks of age, *Mgme1*$^{-/-}$ mice have normal blood hemoglobin concentration, whereas older animals show a mild decrease in hemoglobin at the age of 70 weeks (Supplementary Fig. 4a, b). Importantly, the anemia of *Mgme1*$^{-/-}$ mice is much milder than the anemia in mtDNA mutator mice despite the observation that *Mgme1*$^{-/-}$ mice have much higher levels of subgenomic deleted mtDNA than mtDNA mutator mice[30]. At the age of 70 weeks *Mgme1*$^{-/-}$ mice have moderate anemia without reticulocytosis accompanied by moderate splenomegaly (Supplementary Fig. 4b–d), whereas mtDNA mutator mice at 40 weeks of age have profound anemia, marked reticulocytosis and massive splenomegaly with extramedullary hematopoiesis[25,28].

Finally, we performed combined cytochrome c oxidase/ succinate dehydrogenase (COX/SDH) enzyme histochemistry of heart, colon and skeletal muscle of *Mgme1*$^{-/-}$ mice and controls at the ages of 45 and 70 weeks. We found COX deficient cells in heart and colon from the age of 45 weeks in *Mgme1*$^{-/-}$ mice, whereas there were no changes in skeletal muscle (Fig. 4).

**Increased stability of 7S DNA in the absence of MGME1.** Given the observed effects on mtDNA quantity and integrity in both patients and mice lacking MGME1, we investigated regulation of replication in the control region. Typically, most mtDNA replication events initiated at the O$_H$ region are abortive due to premature termination at the end of the control region, which results

in the formation of a ~650 nt long nascent DNA species (7S DNA) that creates a characteristic triple-stranded DNA structure, the displacement loop (D loop)[31]. Although the smear around 7S DNA in *Mgme1*$^{-/-}$ mice samples prevented us from accurate quantification of the 7S DNA band, Southern blot analyses suggested an increase of steady-state levels of 7S DNA in *Mgme1*$^{-/-}$ mice (Fig. 5a, b). To further investigate the mechanism behind the altered levels of 7S DNA levels, we performed in organello mtDNA replication experiments (Fig. 5c)[32,33]. Freshly isolated mitochondria from heart tissue were pulse labeled for 2 h with $^{32}$P-dATP followed by a one-hour chase to follow the synthesis and stability of de novo synthesized mtDNA (Fig. 5c). Surprisingly, no de novo synthesis of a distinct 7S DNA species was found in heart mitochondria from *Mgme1*$^{-/-}$ mice (Fig. 5c, lanes 2 and 6). However, we observed robust ongoing mtDNA synthesis as there was labeling of full-length mtDNA and a smear of shorter mtDNA species in *Mgme1*$^{-/-}$ mitochondria, similar to the findings in wild-type mitochondria (Fig. 5c). The replicative DNA helicase TWINKLE is essential for mtDNA replication[34], and we, therefore, also analyzed *Twnk* knockout (*Twnk*$^{loxP/loxP}$; +/*Ckmm-cre*) heart mitochondria to ensure that the observed labeling is due to de novo mtDNA synthesis. As expected, no de novo mtDNA replication products were detected in *Twnk* knockout mitochondria (Fig. 5c, lanes 10 and 11).

Moreover, we performed chase experiments and found that 7S DNA is less stable than full-length mtDNA in wild-type mitochondria (Fig. 5c, lanes 1, 5, and 9 vs lanes 4, 8, and 12), consistent with previous observations in cultured cells[35]. To investigate the puzzling discrepancy between the low de novo synthesis and the high steady-state levels of 7S DNA in *Mgme1*$^{-/-}$ mice, we followed the stability of mtDNA with Southern blot analyses after inhibiting mtDNA replication with the chain-terminating nucleotide analog 2′,3′-dideoxycytidine (ddC) in mouse embryonic fibroblasts (MEFs). In the presence of ddC no 7S DNA is observed after three days of treatment. In contrast, 7S DNA is stabilized in the absence of MGME1 and can still be observed after 3 days of ddC treatment

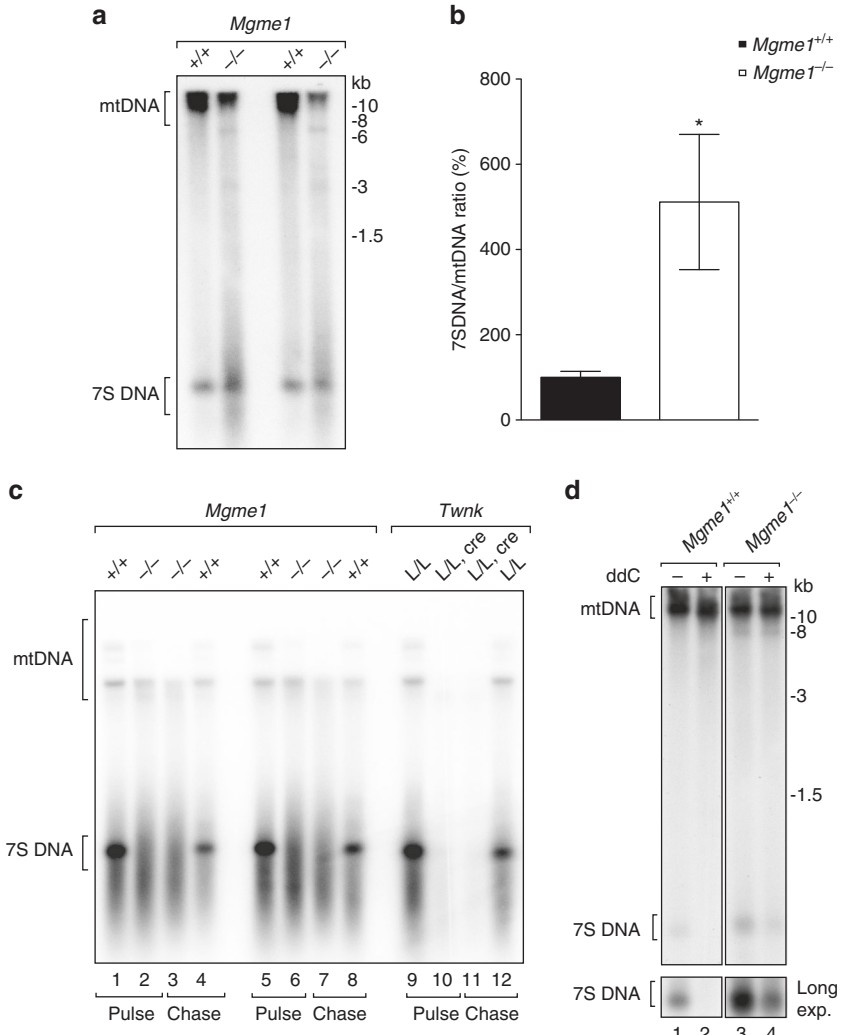

**Fig. 5** Increased 7S DNA stability and diminished 7S de novo synthesis in *Mgme1* knockout mice. **a** Southern blot analysis of SphI-digested mtDNA in heart from wild-type (+/+) and *Mgme1* knockout mice (−/−). **b** Southern blot quantification: ratio of 7S DNA to mtDNA signal. Error bars represent the SEM. *$P < 0,05$; Student's *t*-test. $n = 4$ biological replicates per genotype. **c** De novo DNA synthesis in heart mitochondria isolated from 8-week-old control (*Mgme1*$^{+/+}$) and *Mgme1* knockout (*Mgme1*$^{−/−}$) mice and *Twnk* control (*Twnk*$^{L/L}$) and tissue-specific knockout (*Twnk*$^{L/L}$, *cre*) heart mitochondria. Mitochondria were pulse labeled for 2 h and the chase was performed for 1 h. **d** Southern blot analysis of SphI-digested mtDNA and 7SDNA from 3 days ddC treated (+) or untreated (−) mouse embryonic fibroblasts

(Fig. 5d, lane 4 vs lane 2). To elucidate whether the increased stability of 7S DNA was associated with structural changes, we defined the 3′ and 5′ ends of 7S DNA in *Mgme1*$^{−/−}$ mice (Supplementary Fig. 5a, b). Ligation-mediated (LM)-PCR analysis showed a tendency of shift towards longer 5′ DNA ends in the absence of MGME1 (Supplementary Fig. 5a, b). In contrast, LM-PCR analyses as well as 3′ polyadenylation-mediated PCR amplification revealed only modest changes in the 3′ ends of 7S DNA of *Mgme1*$^{−/−}$ mice (Supplementary Fig. 5a). Extended 5′ ends of 7S DNA have also been reported in fibroblasts from patients with *MGME1* mutations and it has been speculated that this may explain the increased stability of 7S DNA[23].

**MGME1 interacts with mtDNA replication factors**. To further establish a role for MGME1 in mtDNA replication we performed a search for interacting partners by using a proximity-biotinylation assay (BioID)[36] followed by affinity purification and mass spectrometry. The expression of human MGME1-BirA* in cultured human cells leads to an increase of protein biotinylation (Supplementary Fig. 6a) and the BirA tag does not interfere with the

mitochondrial localization of the fusion protein (Supplementary Fig. 6b). Our results confirm that MGME1 interacts with the catalytic subunit of the mtDNA polymerase (POLGA) (Fig. 6)[23,37]. Furthermore, we identified a number of other mitochondrial replication-related proteins that could be interactors of MGME1, e.g., mitochondrial single-stranded DNA-binding protein (SSBP1), mitochondrial RNA polymerase (POLRMT) and TWINKLE (Fig. 6, Supplementary Data 1). As expected, several naturally biotinylated mitochondrial proteins, such as pyruvate carboxylase (PC), propionyl-CoA carboxylase (PCC), and methylcrotonyl-CoA carboxylase (MCCC1) were also identified in the affinity purification experiments. In addition, we identified very abundant mitochondrial proteins, such as proteases and respiratory chain subunits, which likely represent contaminants as we often find these proteins in various types of protein interaction studies.

**Lack of MGME1 influences mitochondrial transcription**. Next, we investigated the steady-state levels of mitochondrial rRNAs, tRNAs, and mRNAs in heart tissue from *Mgme1*$^{−/−}$ mice. Unexpectedly, there was a decrease in the abundance of

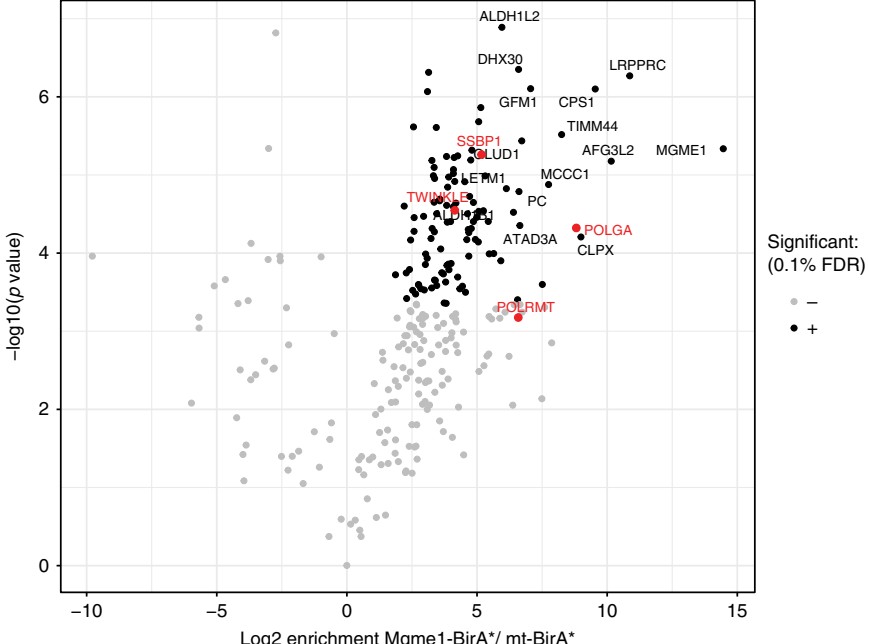

**Fig. 6** MGME1 interacts with components of the mtDNA replication machinery. Volcano plot showing proteins enriched after streptavidin pull-down performed in HeLa cells transfected with MGME1-BirA* upon biotin treatment. Mitochondrial targeted BirA* (mt-BirA*) was used as a control. Enriched mtDNA replication factors are highlighted in red. Gray dots nonsignificant, black and red dots significant hits. Student's *t*-test with adjusted *P*-value (Benjamini–Hochberg false discovery rate-FDR correction of less than 0.1)

mitochondrial transcripts from HSP and an increase of promoter–proximal transcripts from LSP in $Mgme1^{-/-}$ mice (Fig. 7a, b, Supplementary Fig. 7a). Promoter-distal LSP transcripts, such as tRNA$^{Asn}$ and tRNA$^{Cys}$, did not change significantly in $Mgme1^{-/-}$ mice. Normally, there is frequent termination of LSP transcription at CSB1 to generate the abundant polyadenylated 7S RNA (Fig. 7c) with a poorly understood function[38]. The steady-state levels of 7S RNA were markedly decreased in $Mgme1^{-/-}$ mice (Fig. 7c, Supplementary Fig. 7b). In contrast, we found markedly increased levels of an antisense H-strand transcript that spans over the control region, the so called anti control region transcript (ACR) (Fig. 7d). This transcript, which has an opposite sense to LSP transcripts, has previously been reported to increase in abundance in response to decreased mtDNA transcription initiation, decreased mtDNA replication or inhibition of mitochondrial translation[38–40,41]. This finding is in nice agreement with previous results showing that in the absence of premature termination of mtDNA replication (leading to decreased de novo formation of 7S DNA) the levels of the ACR transcript are increased[38,41].

**Loss of MGME1 does not affect mitochondrial protein levels.** As loss of MGME1 leads to decreased levels of full-length mtDNA and the formation of high levels of a linear subgenomic mtDNA molecule (Fig. 2), we decided to investigate the steady-state levels of mitochondrial proteins encoded by mtDNA and nuclear genes (Fig. 8). The steady-state levels of (OXPHOS) subunits were normal in mitochondrial protein extracts (Fig. 8a, b) and the organization of the respiratory complexes was unaltered on blue native polyacrylamide gel electrophoresis (BN-PAGE) of mitochondrial extracts from $Mgme1^{-/-}$ mice (Fig. 8c). We also found normal levels of proteins involved in mtDNA replication, transcription, and RNA maturation, such as mitochondrial transcription termination factor 1 (MTERF1)[42,43], leucine-rich pentatricopeptide repeat-containing (LRPPRC), mitochondrial transcription factor A (TFAM)[44,45], POLRMT[46],

and TWINKLE[34] (Fig. 8a, b). Despite accumulation of linear subgenomic mtDNA, depletion of mtDNA and replication stalling in young animals, there were no clear OXPHOS defects. This is probably due to compensatory mechanisms acting on various levels of the mtDNA expression axis thus resulting in high threshold levels, i.e., the overall mtDNA expression must be substantially reduced before respiratory chain deficiency occurs.

**Tissue-specific replication stalling in $Mgme1$ knockout mice.** To gain a more detailed insight into the integrity of mtDNA we performed next generation sequencing of mtDNA from liver (Fig. 9a), heart (Fig. 9d) and brain (Supplementary Fig. 8) of $Mgme1^{-/-}$ mice. The relative sequence coverage pattern of mtDNA from the liver of $Mgme1^{-/-}$ mice (Fig. 9a) was very similar to the previously published pattern in liver from $PolgA^{mut/mut}$ mice[25,26] and is consistent with the presence of a linear subgenomic mtDNA fragment extending from O$_H$ to O$_L$ in both $Mgme1^{-/-}$ (Fig. 2d, Supplementary Fig. 2d) and $PolgA^{mut/mut}$ mice[25]. Further analysis of mtDNA by neutral-neutral two-dimensional agarose gel electrophoresis (2DNAGE) showed a prominent and site-specific stalling of mtDNA replication in the region of O$_L$ (Fig. 9c) consistent with the sharp decline in sequence coverage after this point. In striking contrast, the sequence coverage pattern of mtDNA from heart (Fig. 9d) and brain (Supplementary Fig. 8a) of $Mgme1^{-/-}$ mice showed more of a gradual decrease with the most abundant sequences being present close to O$_H$. 2DNAGE analysis of heart and brain mtDNA from $Mgme1^{-/-}$ mice showed a generalized increase in the abundance of replication intermediates at all points along the replication fork arc (Fig. 9e, f; Supplementary Fig. 8b, c; Supplementary Fig. 9). This is consistent with a non-specific replication stalling phenotype as documented by the decreased sequence coverage as the distance from the origin of replication increases.

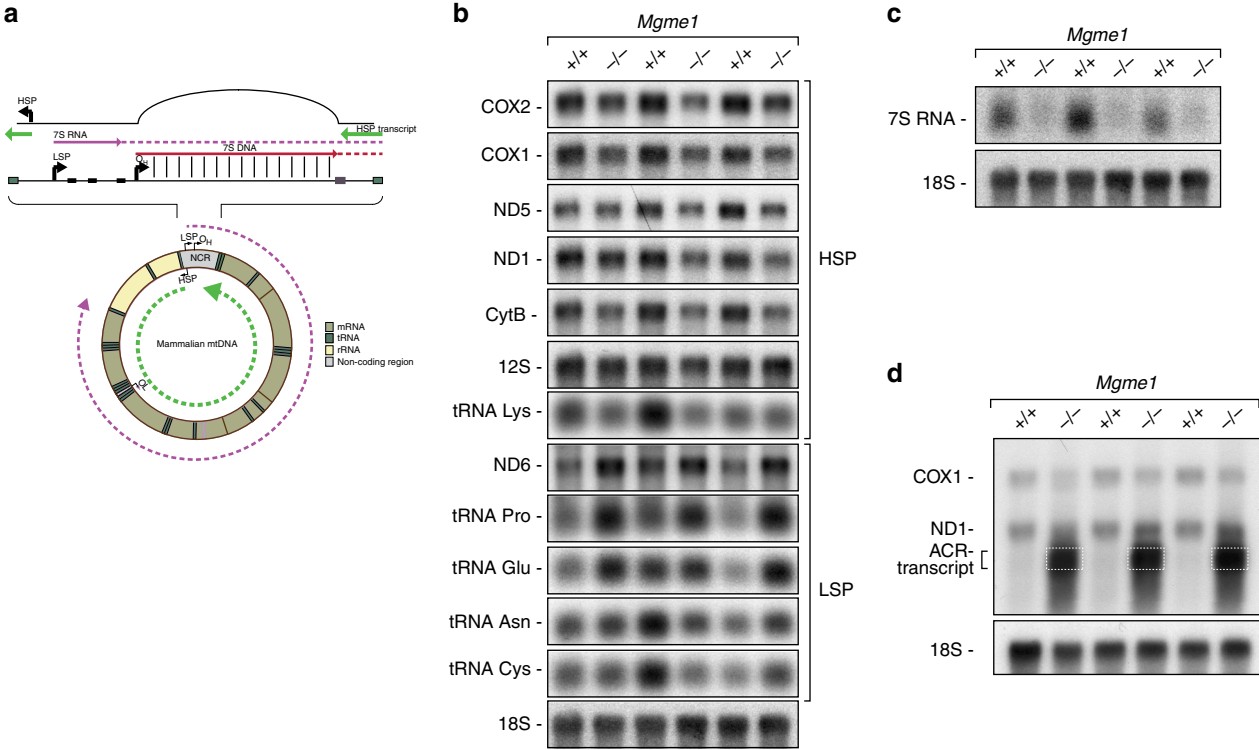

**Fig. 7** Steady-state levels of mitochondrial transcripts in *Mgme1* knockout mice. **a** Schematic representation of mtDNA with focus on NCR (non-coding region). Green line heavy-strand transcripts, pink line light-strand transcripts. Red line heavy-strand replication, HSP heavy-strand transcription promoter, LSP light-strand transcription promoter, $O_H$ origin of heavy-strand replication, $O_L$ origin of light-strand replication **b** Steady-state levels of mitochondrial mRNAs, rRNAs, and tRNAs; Nuclear 18S rRNA is used as a loading control. COX1 (Cytochrome oxidase subunit 1); COX2 (Cytochrome oxidase subunit 2); ND5 (NADH-ubiquinone oxidoreductase chain 5); ND1 (NADH-ubiquinone oxidoreductase chain 1); ND6 (NADH-ubiquinone oxidoreductase chain 6); CytB (Cytochrome reductase subunit b). **c** Northern blot analysis of 7S RNA levels in heart from control (*L/L*) and *Mgme1* tissue-specific knockout mice (*L/ L, cre*) and *Mgme1* knockout mice (−/−) and corresponding control (+/+); loading control is nuclear 18S rRNA. Figures **b** and **c** are derived from the same northern blot membrane. **d** The ACR transcript mapped by northern blot analysis, using single-stranded riboprobe complementary to the mtDNA light-strand control region. ND1 and COX1 transcripts were used as size indicators and 18S nuclear rRNA was used as a loading control. ACR transcript is indicated by dotted box

## Disscussion

In this study, we report that the loss of MGME1 has marked effects on mtDNA replication in vivo. The $Mgme1^{-/-}$ mice are viable but exhibit an mtDNA replication stalling phenotype that surprisingly shows a tissue-specific pattern and is associated with decreased mtDNA copy number and an accumulation of mtDNA deletions. Furthermore, we report an unexpected role for MGME1 in the regulation of mtDNA replication and transcription termination at the end of the D-loop region. Proximity labeling experiments with the Bio-ID method showed that MGME1 interacts with a number of components of the replication machinery. MGME1, thus, has an important role in the regulation of mtDNA replication.

Loss of MGME1 in mice causes an increase in the steady-state levels of 7S DNA, consistent with the previously suggested role of MGME1 in degrading this mtDNA species[7,23]. Importantly, de novo replication of mtDNA occurs in the absence of MGME1, but the formation of 7S DNA is severely decreased. This striking discrepancy between impaired de novo synthesis and increased steady-state levels of 7S DNA is likely explained by increased stability of these molecules in the absence of MGME1 activity. It has been estimated that the 7S DNA in patient fibroblasts has a four-fold longer half-life than in control cells[23], and here we report increased stability of the 7S DNA species in $Mgme1^{-/-}$ MEFs after mtDNA replication inhibition with the chain-terminating nucleotide analog ddC. Interestingly, the severely decreased de novo formation of 7S DNA argues that MGME1

may also modulate replication termination at the end of the D-loop region. Support for a regulatory role for MGME1 at the end of the D-loop region comes from the observation that transcription from HSP, which is normally terminated at this region, is aberrantly regulated in the absence of MGME1. Lack of HSP transcription termination results in accumulation of an ACR transcript over the D-loop region. This ACR transcript has also been described to form in response to thiamphenicol or ddC treatment of tissue culture cells[38,40]. We have previously reported in vivo evidence for ACR transcript accumulation in transgenic mice with impaired mtDNA transcription initiation[41]. In this mouse strain, knockout of *Tfam* in heart, which typically results in severe cardiomyopathy, was rescued by expression of the human *TFAM* gene. Interestingly, in addition to ACR transcript accumulation in those rescue mice the levels of 7S DNA were significantly reduced. We propose that MGME1 might be part of regulatory switch acting at the end of the D-loop region that controls mtDNA replication and H-strand transcription termination. Similarly, we have recently shown that the mitochondrial transcription termination factor 1 (MTERF1) blocks transcription initiated from LSP, just downstream of the ribosomal transcription unit, preventing formation of an antisense transcript over the D-loop region and interference with the activity of the LSP promoter in the mouse[42]. Moreover, MTERF1 was recently suggested to display contrahelicase activity and would therefore be able to counteract DNA unwinding by the TWINKLE helicase, which is a termination mechanism that may prevent the

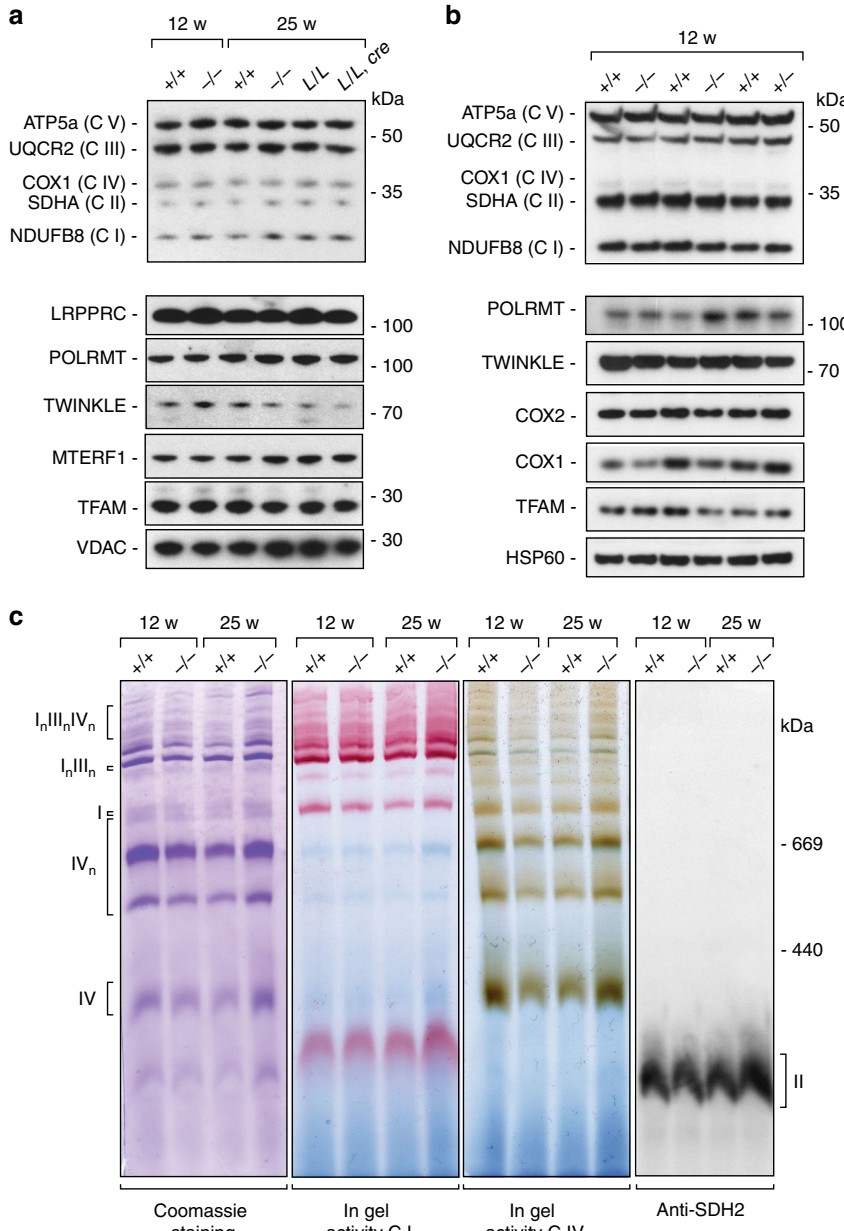

**Fig. 8** Protein and respiratory complexes steady-state levels in *Mgme1* knockout mice. **a** Western blot analysis of steady-state levels of respiratory chain complex subunits and diverse mitochondrial proteins in heart mitochondrial extracts from 12 and 25-week-old control (+/+) and *Mgme1* knockout mice (−/−) mice and heart from control (L/L) and *Mgme1* tissue-specific knockout mice (L/L, cre). **b** Analysis performed as in (**a**) using liver mitochondria isolated from control (+/+), *Mgme1* heterozygous (+/−), and *Mgme1* knockout mice (−/−). **c** BN-PAGE analysis of *Mgme1* knockout (−/−) and wild-type (+/+) heart mitochondria. Left panel, coomassie staining. Middle pannels, in-gel enzyme activities of complexes I and IV. Right panel, western blot analysis of steady-state levels of respiratory chain complex II visualized by immunostaining against SDH2 subunit. $I_nIII_n$ and $I_nIII_nIV_n$: respiratory chain supercomplexes

replication and transcription machineries from colliding[43]. Interestingly, ChIP sequencing experiments recently demonstrated that TWINKLE may also be important for the regulation of replication at the 3′ end of 7S DNA in the control region[38]. Importantly, the Bio-ID experiments we present here show an interaction between MGME1 and TWINKLE. The prevention of interference between the mitochondrial replication and transcription machineries, as well as prevention of promoter interference seems to require complex regulation on multiple levels.

In line with the above-discussed results, *Mgme1*$^{-/-}$ mice display reduced steady-state levels of the transcript originating from the HSP promoter, whereas promoter–proximal transcripts from

LSP are increased and promoter-distal transcripts unaltered. It is likely that the ACR transcript, if abundant, interferes with the HSP promoter leading to a decrease of HSP transcription initiation. As a consequence, the transcription apparatus would shift to the LSP promoter thus resulting in an increased transcription initiation from LSP and higher levels of LSP promoter–proximal transcripts. Interestingly, despite boosted proximal transcription from LSP, the 7S RNA levels were decreased in *Mgme1*$^{-/-}$ mice. The function of this promoter proximal transcript is unknown, but it is often discussed in the context of replication primer formation[38,47]. We propose that most LSP initiation events are used for DNA synthesis in our

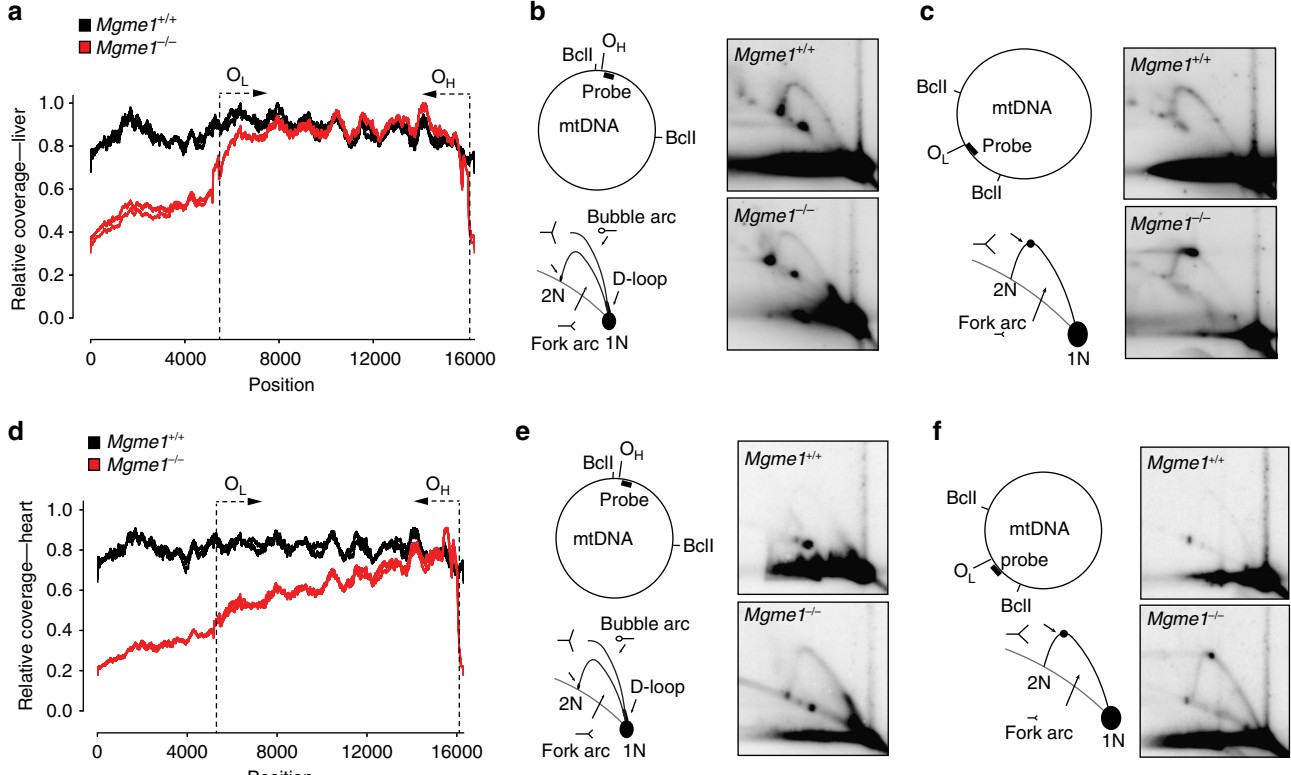

**Fig. 9** Replication stalling in heart and liver of *Mgme1* knockout mice visualized by next generation sequencing and 2DNAGE. **a** Sequence coverage of the mouse mtDNA samples from liver of *Mgme1*$^{-/-}$ and controls. Mitochondrial genome position (*x*-axis) versus sequence coverage divided by maximum coverage for each sample. For each genotype two samples derived from different mice were analyzed. The approximate locations of the origins of light-strand (O$_L$) and heavy-strand (O$_H$) replication are indicated by dotted lines with arrows. **b** and **c** mtDNA replication pattern in the liver of *Mgme1*$^{+/+}$ and *Mgme1*$^{-/-}$ mice analyzed by two-dimensional neutral-agarose gel electrophoresis followed by Southern blot. Restriction enzymes and probes used are indicated to the left. The black bars indicate probe used, and O$_H$ marks the origin of H-strand replication and O$_L$ denotes the origin of L-strand replication. **d** Same as (**a**) except that mtDNA isolated from heart tissue was used. **e**, **f** Analysis performed as in (**b**) and (**c**) but with heart samples. 1N indicates the migration of a fragment of non-replicating mtDNA, and 2N indicates the migration of fully-replicated mtDNA

experimental model, thereby leading to the absence of 7S RNA. In agreement with this model, we report increased sequence coverage reads consistent with a replication initiation boost at O$_H$ and an increase of promoter-proximal LSP transcripts in *Mgme1*$^{-/-}$ mice.

Lack of MGME1 causes mtDNA depletion and accumulation of mtDNA deletions in a range of different mouse tissues. A hallmark of MGME1 deficiency in patient fibroblasts and mice is an 11 kb linear mtDNA fragment spanning the entire major arc of the mtDNA, which has been previously described in mtDNA mutator mice[25] and flies[48]. According to a recently suggested model, the 11 kb linear abnormal replication products arise from failed ligation of the nascent H-strand at O$_H$ and are a consequence of non-ligatable flaps created at the origin of replication[27,49]. Numerous studies suggest that mtDNA mutations and deletions contribute to the ageing phenotypes in experimental animals and in humans[13,50–52]. Indeed, mtDNA mutator mice develop progressive premature ageing syndrome phenotypes[25,53]. In addition to the presence of the above mentioned 11 kb subgenomic mtDNA species, the mtDNA mutator mice also accumulate an increased number of point mutations, that most likely drive the ageing phenotype[54]. Consistent with this hypothesis, *Mgme1*$^{-/-}$ mice do not accumulate point mutations and do not display a progeroid phenotype. In line with this finding, mtDNA subgenomic fragments have not been detected in tissues from aging mammals further indicating that this lesion on its own does not induce ageing[51].

Both 2DNAGE and sequence analysis of mtDNA isolated from various *Mgme1*$^{-/-}$ tissues revealed remarkable tissue-specific molecular phenotypes. The 2DNAGE analysis of *Mgme1*$^{-/-}$ heart and brain showed a general and unspecific accumulation of replication intermediates along the replication arc, suggesting that replication is stalling along the entire restriction fragment. Similarly, downregulation of MGME1 in tissue culture cells induces a comparable non-specific stalling effect[7]. This result was different to what was seen in liver tissue of the *Mgme1*$^{-/-}$ mice, where a very prominent stalling site was observed in the vicinity of O$_L$. Furthermore, deep sequencing of liver mtDNA from *Mgme1*$^{-/-}$ mice showed a sequence coverage plot that reflects the presence of linear fragments with a large deletion. A similar sequence coverage has been previously reported from liver samples of mtDNA mutator mice[26]. The coverage profiles of *Mgme1*$^{-/-}$ samples from brain and heart displayed a different pattern including a pronounced peak of 7S DNA sequence and loss of sequence coverage in the direction of replication. These differences between the liver on the one hand and heart tissue on the other hand are in agreement with the distinct replication stalling profiles revealed by 2DNAGE data from these tissues. The increased reads along the major arc in the liver indicate that replication has trouble passing O$_L$, predominantly replicating the major arc, explaining the specific stalling in the 2DNAGE. The unusual slope in sequence coverage from the heart suggests that replication is initiated faster than it can be completed, resulting in more reads closer to the origin and generalized stalling as seen on the 2D gel experiments.

The observed tissue specificity of our $Mgme1^{-/-}$ mouse model is in accord with the well-known tissue heterogeneity of mitochondrial disorders[55]. The tissue-specific phenotypes may, at least partly, be explained by the observation that diverse tissues have different energy demands and biosynthetic capacities. The synthesis of mtDNA relies tightly on a balanced dNTP pool, which depends on the mitochondrial dNTP salvage pathways and assisted transport of cytosolic dNTPs to mitochondrial matrix[56]. These processes are likely differentially regulated in various tissues, thereby contributing to the tissue-specific manifestations of mitochondrial mtDNA maintenance diseases as exemplified by patients carrying mutations in enzymes important for the mitochondrial nucleotide salvage pathways, e.g., TK2[10] and MPV17[57].

In conclusion, we show here that MGME1 is a replication-related nuclease, necessary for 5′ 7S DNA processing and faithful mtDNA replication in mice. Absence of MGME1 results in tissue-specific replication stalling and accumulation of deleted mtDNA molecules but does not create a progeroid phenotype in affected animals. We propose that MGME1 is part of a termination complex acting at the end of the D-loop region where it modulates mtDNA replication and H-strand transcription termination. The tissue-specific molecular phenotypes associated with MGME1 deficiency suggest that the $Mgme1^{-/-}$ mice will be a valuable model for investigation of the pathophysiology of mtDNA maintenance disorders.

## Methods

**Animals and housing**. Knockout transgenic mice on a C57BL/6N background were housed in standard individually ventilated cages ($45 \times 29 \times 12$ cm) under a 12 h light/ dark schedule in controlled environmental conditions of $22 \pm 2$ °C and 50 + 10% relative humidity and fed a normal chow diet and water ad libitum. The study was approved by the Landesamt für Natur, Umwelt und Verbraucherschutz Nordrhein–Westfalen (reference numbers 84-02.04.2015.A103 and 84-02.50.15.004) and performed in accordance with the recommendations and guidelines of the Federation of European Laboratory Animal Science Associations (FELASA).

**Generation of *Mgme1* knockout mice**. The targeting vector for disruption of *Mgme1* in embryonic stem cells has been generated using BAC clones from the C57BL/6J RPCIB-731 BAC library and has been transfected into the Taconic Artemis C57BL/6N Tac ES cell line. To generate conditional knockout *Mgme1* mice, exon III was flanked by loxP sites. The puromycin resistance cassette was introduced as a selection marker and removed by mating of $Mgme1^{+/loxP\text{-}pur}$ mice with transgenic mice ubiquitously expressing *Flp*-recombinase. $Mgme1^{+/loxP}$ mice were mated with mice ubiquitously expressing *cre*-recombinase (β-actin-*cre*) to generate heterozygous knockout $Mgme1^{+/-}$ mice. $Mgme1^{+/-}$ mice were further intercrossed to generate homozygous knockout $Mgme1^{+/-}$ mice. To obtain tissue-specific (heart and skeletal muscle) knockout mice[34] $Mgme1^{loxP/loxP}$ mice were crossed with transgenic mice expressing *cre*-recombinase under the control of the muscle creatinine kinase promoter (*Ckmm-cre*).

**Isolation of mitochondria from mouse tissue**. Mitochondria were isolated from mouse tissues using differential centrifugation as previously described[34]. Briefly, fresh tissues were cut, washed with ice cold PBS and homogenized in mitochondrial isolation buffer (MIB) containing 310 mM sucrose, 10 mM Tris-HCl, and 1 mM EDTA by using a Potter S pestle (Sartorius). The homogenate was centrifuged at $1000 \times g$ for 10 min at 4 °C and the supernatant was subsequently spun at $10,000 \times g$ for 15 min at 4 °C to isolate mitochondria. Crude mitochondrial pellets were suspended in MIB supplemented with $1 \times$ Complete protease inhibitor cocktail (Roche). Protein concentration was determined by the Bradford method using BSA as a standard. Mitochondria from brain tissue were isolated following protocol from Miltenyi Biotec mitochondrial extraction tissue kit using TOM22 MicroBeads.

**DNA extraction and Southern blot analysis**. Genomic DNA was isolated by Gentra Puregene Tissue Kit (Qiagen) according to kit instuctions. DNA quantification was performed with the Qubit 1.0 fluorometer (Thermofisher). A volume of 2 μg of DNA were digested with SacI, XhoI, SphI, or EagI and DNA fragments were separated by agarose gel electrophoresis, transferred to nitrocellulose membranes (Hybond-N + membranes, GE Healthcare) and hybridization was carried out with $\alpha^{32}$P-dCTP-labeled probes. For 7S DNA Southern samples were heated for 3 min at 93 °C prior to loading. List of probes can be find in Supplementary Data 2.

Uncropped scans of important Southern blots with marker lanes are presented in Supplementary Fig. 10.

**In organello replication**. A volume of 1 mg of freshly isolated heart mitochondria was resuspended in 0.5 ml of incubation buffer (25 mM sucrose, 75 mM sorbitol, 100 mM KCl, 10 mM $K_2HPO_4$, 0.05 mM EDTA, 5 mM $MgCl_2$, 1 mM ADP, 10 mM glutamate, 2.5 mM malate, 10 mM Tris–HCl, pH 7.4) supplemented with 1 mg/ml fatty acid-free bovine serum albumin, 50 μM each of dTTP, dCTP, and dGTP and 20 μCi $\alpha$-$^{32}$P-dATP (3000 Ci/mmol). Incubation was carried out at 37 °C for 2 h on a rotating wheel. Chase experiments were followed by incubation with non-radiolabeled dATP (5 mM) for additional hour. After incubation, mitochondria were pelleted at 9000 rpm for 4 min and washed twice with 10% glycerol, 10 mM Tris–HCl, pH 6.8, 0.15 mM $MgCl_2$. In the following step DNA isolation and Southern blot analysis were performed as described above.

**Long-extension PCR**. Mouse mtDNA was amplified from 2 ng of total DNA with the following primers (P1:488-510, P2:4021-4040) using LA Taq polymerase (TAKARA, Japan) and following PCR conditions: 98 °C for 10 s, 58 °C for 30 s, and 60 °C for 10 min, 35 cycles.

**Northern blot analysis**. Northern blot transcript analysis was performed as previously described[34]. Briefly, RNA, 2 μg, was isolated using the ToTALLY RNA Total RNA isolation kit (Ambion), separated in formaldehyde agarose gels and transferred to Hybond-N + membranes (GE Healthcare) by northern blotting. DNA probes, $\alpha$-$^{32}$P-dCTP-labeled, were used for visualization of mRNA and rRNA levels. Different tRNAs and 7S RNA were detected using specific oligonucleotides labeled with $\gamma$-$^{32}$P-ATP. For the detection of the ACR transcript riboprobe was synthetized using Riboprobe System T7 Kit (Promega). List of probes can be find in Supplementary Data 2.

**Western blot analysis and BN-PAGE**. A volume of 20 μg of isolated mitochondria were resuspended in $4 \times$ Lämmli-Buffer (4% SDS, 20% Glycerol, 120 mM Tris, 0,02% Bromophenol Blue), proteins were separated on 4–12% NuPage gels (Invitrogen) and transferred on $Hybond^{TM}$-P membrane (GE Helthcare). Primary antibodies used for western blotting were as follows: MitoProfile total OXPHOS antibody cocktail (MitoSciences), HSP60 (Cell signalling, 1:1000), COX1 (Invitrogen, 1:1000), MTERF1 (Proteintech, 1:1000), TFAM (Abonva, 1:1000), SDH2 (MitoScience). Rabbit polyclonal antisera against MGME1, TWINKLE, POLRMT, COX2, LRPPRC proteins were generated from recombinant mouse proteins.

For BN-PAGE 75 μg mitochondria were solubilized in solubilization buffer: 1% (w/v) digitonin (Calbiochem), 20 mM Tris, pH 7.4, 0.1 mM EDTA, 50 mM NaCl, 10% (v/v) glycerol. Following 15 min of incubation on ice, non-solubilized material was removed by centrifugation and the supernatant was mixed with loading dye (5% (w/v) Coomassie Brilliant Blue G-250 (Serva), 100 mM Tris, pH 7, 500 mM 6-aminocaproic acid). Samples were resolved on 3–13% (w/v) acrylamide gradient BN-PAGE gels. BN gels were further subjected to Coomassie Brilliant Blue R staining, in-gel activity assay[58] or western blot analysis, as indicated.

**Library preparation and Illumina pair-end DNA sequencing**. Pair-end sequencing was performed by the Max Planck-Genome-centre Cologne, Germany (http://mpgc.mpipz.mpg.de/home/). mtDNA Quality control was done using the Agilent TapeStation Genomic DNA ScreenTape and the Qubit BR kit. The library preparation protocol/kit was the NEBNext Ultra™ DNA Library Prep Kit for Illumina (NEB) after fragmentation of 150 ng gDNA with the Covaris sonicator to the requested insert size of >500 bp. The sequencing run conditions were $2 \times 250$ bp on the Illumina HiSeq2500, using HiSeq Rapid v2 Kits from Illumina. Each library was indexed individually with the provided barcodes and sequenced to 6,000,000 reads (tolerance range −30%) The reads were aligned to the C57Bl/6 J mouse mtDNA reference sequence (NC_005089.1), using the corona lite mapping algorithm (Applied Biosystems) with default settings. The first 49 bases of the mtDNA sequence were appended to the end of the reference to avoid that reads fail to align due to the circularity of the mitochondrial genome. This alignment procedure attempts to map each read at full-length to the reference sequence, allowing for at most 6 mismatches for each 50 bp read.

**2DNAGE**. For two-dimensional gels, mtDNA was isolated from fresh sucrose gradient purified mitochondria from liver and heart tissues and the by sequential phenol-chloroform extraction. The resulting DNA (3 μg per panel) was digested with BclI, precipitated and loaded onto 0.4% agarose without ethidium bromide. First dimension gels were separated at 27 V for 18 h at room temperature, then DNA-containing lanes were excised with a scalpel, rotated 90° counterclockwise, and molten 1% agarose containing 500 ng/ml ethidium bromide was cast around the gel slices. Second-dimension gels were run at a constant 260 mA for 6 h at 4 °C. Gels were Southern blotted onto nylon membranes and hybridized with probes detecting either the $O_H$-containing fragment or the $O_L$-containing fragment. Primer sequences used for probe synthesis (5′-3′) were: $O_H$ forward, ATCAATGGTTCAGGTCATAAAATAATCATCAAC; $O_H$ reverse,

GCCTTAGGTGATTGGGTTTTGC; $O_L$ forward, TGACTTGTCCCACTAA-TAATCGGAG; and $O_L$ reverse, CCCAAAGAATCAGAACAGATGCTG.

**ddC treatment of mouse embryonic fibroblasts**. 2,3′ Dideoxycytidine (Sigma) was used at a final concentration of 20 μM. Cells were treated for 3 days and afterwards resuspended in fresh medium at a concentration of $1.5–2 \times 10^5$ cells/ml. Total DNA was isolated by Gentra Puregene Tissue Kit and extraction was performed as instructed in the kit.

**Bio-ID**. MGME1-BirA*-HA and MTS-BirA*-HA (MTS from human COX8a) and were cloned into pcDNA3.1 MCS-BirA (R118G)-HA vector (Addgene, Cambridge, MA, USA). Pull-down experiments were performed according to the published protocol[36] with small modifications. HeLa cells were transfected using Lipofectamine 2000 (Invitrogen), treated with 50 μM biotin 24 h after transfection and lysed after 6 h in 2.4 ml lysis buffer (50 mM Tris-Cl, pH 7.4, 500 mM NaCl, 0.2% SDS, 1 × protease inhibitor (Halt Protease Inhibitor Cocktail, EDTA-free, Thermo Fisher Scientific), 1 mM DTT. Subsequently, 240 μl of 20% Triton-X 100 (final concentration 2%) was added. After sonication samples were centrifuged at 16,500× g, 4 °C for 10 min. Dynabeads MyOne Streptavidin C1 (Thermo Fisher Scientific) were incubated with the lysates on a rotator at 4 °C over night. Following washing in 1.5 ml wash buffer 2 (0.1% (w/v) deoxycholic acid, 1% (w/v) TritonX-100, 1 mM EDTA, 500 mM NaCl, 50 mM HEPES, pH 7,5) five successive washing steps with 1,5 ml 50 mM Tris-Cl, pH 7.4, were applied. Samples were eluted in elution buffer (2 M Urea, 5 ng/μl Trypsin, 1 mM TCEP, 50 mM Tris-Cl, pH 7.5) at room temperature. A volume of 5 mM CAA was added to the samples and reaction was incubated at 37 °C over night. The samples were further analyzed by LC MS/MS mass spectrometry. The mass spectrometry proteomics data have been deposited to the ProteomeXchange Consortium via the PRIDE partner repository with the dataset identifier PXD009138.

**COX/SDH double-labeling enzyme histochemistry**. COX/SDH double staining was performed as previously described[29]. Briefly, fresh heart and skeletal muscle tissues were dissected and immediately frozen in isopentane chilled with liquid nitrogen. Colon tissue was upon dissection frozen with liquid nitrogen. Tissues were further cryosectioned into sections (10 μm for skeletal muscle and colon; 7 μm for heart), mounted on slides and left to air dry briefly. Freshly prepared buffer A (0.8 ml of 5 mM 3,3′-diaminobenzidine tetrahydrochloride, 0.2 ml of 500 μM cytochrome *c* and 10 μl of catalase) was added to the slides. After incubation for 60 minutes at 37 °C, slides were washed three times by 0.1 M phosphate buffered saline, pH 7.0. Then freshly prepared buffer B (0.8 ml 1.875 mM of nitroblue tetrazolium, 0.1 ml 1.3 M of sodium succinate, 0.1 ml 2.0 mM phenazine methosulphate, and 10 μl of 100 mM sodium azide) was applied and incubated for 30 minutes at 37 °C. Slides were washed three times with 0.1 M phosphate buffered saline, pH 7.0, dehydrated and mounted for bright-field microscopy.

**Sperm motility analysis**. Sperm motility analysis was performed using Computer-assisted semen analysis (CASA) as previously described[29]. Briefly, the cauda region of the right epididymis of each mouse was clamped proximally and distally, excised and rinsed briefly in pre-warmed PBS, and placed in a 1.5 ml Eppendorf tube containing fresh, pre-warmed M2 medium (SIGMA). The cauda epididymis was then unclamped and pierced with the point of a scalpel blade to allow sperm to diffuse into the medium. Sperm were allowed to disperse for 10 min at 37 °C and were then appropriately diluted with fresh medium to permit sperm motility analysis of 10 μL sperm suspension by CASA detection (Hamilton Thorne Research Inc. Beverly, MA, USA).

**Data availability**. All the data needed to evaluate the conclusions in the paper are present in the paper and/or the Supplementary Information. The Illumina sequencing data is data freely available in the NCBI Sequence Read Archive under the Study ID SRP132950 under Accession Numbers SRX3712521—SRX3712536 (https://www.ncbi.nlm.nih.gov/Traces/study/?acc=SRP132950)[http://www.ncbi.nlm.nih.gov/bioproject/434306]. The additional data related to this paper may be requested from the authors.

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

## Acknowledgements

This study was supported by Max Planck Society, the Swedish Research Council (2015-00418) and the Knut and Alice Wallenberg foundation to N.G.L. A.F. receives support from the Australian Research Council (DP170103000) and the National Health and Medical Research Council of Australia (APP1067837 and APP1058442). O.R. receives support from the Cancer Council Western Australia. We are grateful to Petra Kirschner and Regina Dirksen for expert technical help. We thank the Max Planck-Genome-centre Cologne (http://mpgc.mpipz.mpg.de/home/) for performing pair-end sequencing in this study. We thank the Bioinformatics Core Facility of the Max Planck Institute for Biology of Ageing for help with sequence analysis.

## Author contributions

S.M. performed experimental work, data analysis, and was involved in project planning. M.J., T.J.N., J.P.U., C.D.-S., M.-L.S., O.R., A.F., P.L.P., J.B.S., and D.M., helped with experimental work and were involved in project planning and data analysis. X.L. and I.A. analyzed MS data. M.F., A.F., O.R., and J.B.S. analyzed the data and discussed the results. D.M. and N.-G.L. conceived the project and wrote the manuscript.

## Additional information

**Competing interests:** The authors declare no competing interests.

