## [Peer Review File · Nature Communications]

Reviewers' comments:

Reviewer #1 (Expert in transcription/replication; Remarks to the Author):

The manuscript entitled "Tissue-specific patterns of mtDNA replication stalling in mice lacking MGME1" by Matic et al., with Figures 1-7, Supplementary Figures 1-4 and Supplementary Table 1, describe the molecular phenotypes of MGME1 deletion in mice. They generated MGME1 knockout mice and heart and skeletal muscle-specific knockout mice and analyzed the consequences of deletion of the protein on mtDNA, 7S DNA, mitochondrial transcripts and mitochondrial proteins. The authors also analyzed the replication intermediates of mtDNA and attempted to find interacting proteins with MGME1. Although the authors performed many different experiments, I have numbers of concerns considering the standards of Nature Communications. The main concerns are described below.

(1) Similar findings regarding the effect of MGME1 deletion on mtDNA and 7S DNA were already reported in the preceding works using patient fibroblasts having MGME1 null mutation and MGME1-siRNA knockdown cells (Kornblum et al Nature Genet 2013, Nicholls et al Hum Mol Genet 2014), such as multiple deletions of mtDNA, appearance of the 11 kb sub-genomic fragment, elevated levels of 7S DNA, reduced rate of 7S DNA decay, defects in the processing of 5' termini of 7S DNA and an accumulation of replication intermediates which is similar to the results of mouse liver mtDNA in this manuscript.

(2) The authors concluded that the long linear deleted mtDNA species is not the cause of premature ageing phenotype of mtDNA mutator mice based on the observation that MGME1 knockout mice have the linear species but do not show any ageing phenotypes. I understand that this is one of the main conclusions that this manuscript wishes to put forward strongly, as it is clearly stated in SUMMARY and DISCUSSION. However, no data regarding the physiological aspects of the MGME1 knockout mice are presented.

(3) Throughout the manuscript the authors discussed about the increase/decrease of bands of DNA, RNA and proteins. However, they were not supported by the quantification data.

(4) The authors observed difference in the replication intermediates of mtDNA from two organs, liver and heart of the knockout mice using 2DNAGE. The difference is interesting. A strong spot at O(L) was observed in the liver mtDNA, but not in the heart mtDNA, of the mutant mice. However, why such differences are caused was not addressed.

(5) In SUMMARY, the authors stated, "we also report a role for MGME1 in the regulation of replication and transcription termination at the end of the control region of mtDNA". I am afraid it is an overstatement. They analyzed 7S DNA, transcripts in the non-coding region of mtDNA and mitochondrial transcripts. Then, from the data they speculated the possible role of MGME1, but not demonstrated it.

(6) I am afraid that it is difficult to follow and understand the content of the manuscript as the description (explanation) of the data are not sufficient in the text and legends to figures.

(7) The whole set of data that were obtained from the knockout mice would be helpful for other researchers who are interested in MGME1 proteins. However, since the protein has been already studied both in vitro and in living cells extensively, the impact of this work to the relevant field is not strong enough to be considered in Nature Communications.

Overall, I am afraid but this manuscript does not meet the standards of Nature Communications. I

attached my specific comments below with a hope of helping the authors to prepare a new manuscript elsewhere.

Specific comments

Lines 60-63

I am afraid that the content of the sentence is ambiguous. At least a couple of 'extensive in vitro work' should be described to help readers' understanding. Furthermore, a number of animal models, not only mouse but also other species were already generated and intensively studied in the past and many important insights were gained. Thus, it would be helpful if such preceding animal studies are introduced briefly here as a comparison of the current work.

Line 70

Is the reference correct?

Lines 92-98

"homozygous knockout (Mgme1^{-/-}) mice that appeared healthy when followed until the age of 12 months."

"... heart- and skeletal-muscle-specific Mgme1 knockout mice (Mgme1^{loxP/loxP}; +/-Ckmm-cre) that had no obvious phenotype when followed until 12 months of age."

One of the conclusions which this manuscript put forward strongly is that knockout of MGME1 does not cause any ageing phenotype. However, no data related to it is shown. If the authors wish to make such a proposition, data that support the statements should be sufficiently presented. As the authors mentioned "animal models are nevertheless essential to understand the in vivo metabolic consequences" in Introduction, the strong point of animal model studies is that they enable researchers to study physiological and behavioral aspects of the gene of interest using the animals. Therefore, such data are expected to be presented in this manuscript.

Furthermore, discussion regarding the similarity/difference of the phenotypes of the mice and patients with Mgme1 mutations is lacking. I believe it is another strong point of animal model studies and thus such discussion should be made.

Lines 77-78

"MGME1 is not essential for mouse embryonic development"

In relation to above, data supporting the statement should be presented, such as no significant difference in the birth rate of +/+ and -/- ?

Line 95 and Figure 1c

In Figure 1c, how the band indicated by a line was confirmed to be MGME1? Convincing data should be presented to demonstrate the identity of the band. This is important as another band which is migrated slightly faster is also clearly detected by the antibody against MGME1. For example, siRNA (more than one siRNA should be used) knockdown of MGME1 gene in cultured mouse cells would be a good and straightforward experiment to confirm the identity of the band.

Evidence of successful knockout in other organs than heart in the -/- mice needs to be shown by similar western blotting. Sufficient data to demonstrate the successful tissue-specific knockout in heart and skeletal muscle specific knockout mice should be also shown.

Lines 101-102

Readers will find it difficult to understand how the gel image of Fig. 2a demonstrate multiple deletion

of mtDNA in $-/-$ samples as it is not explained. Proper explanation of the data should be described in the text or the legend to the figure.

Line 104

“showed both mtDNA depletion... (Fig. 2b,c)”

To confirm the authors' conclusion, quantification of the band should be shown as graphs with sufficient number of samples with statistical analysis.

Provision of quantification data with statistical analysis should apply to all other data sets with which the authors wish to discuss about increase/decrease and make quantitative interpretation/discussion. In addition, quantification of the comparison of the band intensity of the intact mtDNA and ~11 kb in MGME1 KO samples (and polg mut/mut samples) would be informative when readers consider similarity/difference between MGMT KO mice and polg mut/mut mice.

Fig. 2b,c and S. Fig. 1b, c

How the sizes of the bands were identified so accurately (7.1 and 3.8 and 8.4 and 2.6 kb indicated at the left side of Southern blotting images in Fig. 2 and S Fig. 1b, respectively)? It would be informative if such information is added; for example, DNA ladder (size marker) that was electrophoresed with the samples on the same gels should be shown.

In lane 6 in Fig. 2b, the molecular sizes of the two bands with asterisks appear to be different from those in lanes 2 and 4. Are they the same species?

S. Fig. 1c

The lower portion of the gel image needs to be shown in the similar way to S. Fig. 1b to demonstrate the absence of any smaller bands in $-/-$ samples.

Lines 115-116

“The linear deleted mtDNA molecules are very similar in size”

No appreciable bands were seen in mut/mut PolgA sample lanes in Fig. 2b and S. Fig. 1b. Longer exposure images should be shown.

Lines 127-128

“By using Southern blot analyses, we found increased steady-state levels of 7S DNA in Mgmt1 $-/-$ mice (Fig. 3a).”

I am afraid that in Fig. 3a 7S DNA itself does not appear to increase significantly. The apparent increase of the band intensity of 7S DNA may be an effect of the stronger background smear in $-/-$ lanes. To eliminate the ambiguity, it is suggested that a DNA probe which does not give such smear is used and then the quantification data are provided. Or, sufficient explanation should be provided how the authors came to their conclusion with the current image.

Lines 156-170

The BioID would be a powerful tool to identify proteins that interact with the protein (in this case MGME1) fused with biotin ligase in living cells. However, since the fusion protein is expressed exogenously (and presumably overexpressed), it may be possible that some proteins that are not interacted with MGME1 under physiological conditions are captured under BioID conditions. Thus, if authors wish to put forward the conclusion that POLRMT, TWINKLE, SSPB1 and POLGA interact with MGME1, it is required that the interactions should be supported with another experiment, such as CoIP. Otherwise, the authors should rewrite the paragraph carefully with consideration of how far one can say with BioID only. This applies to all the sentences which is related to the data elsewhere in the manuscript.

Lines 172-183.

It is necessary to show the quantification graphs of all RNAs which are discussed in the paragraph. In Figure 5c, the lower band is indicated as 7S RNA. Then what is the upper band? Explanation and/or supporting data on how the authors can know the indicated band is 7S RNA would be necessary. It would be helpful if ACR transcript is indicated clearly in Figure 5a. Here, most readers would not be able to follow the authors' argument as ACR transcript is not sufficiently explained.

Lines 182-183.

"This finding is in nice agreement with previous results that in the absence of premature mtDNA replication termination (i.e. 7S DNA) the levels of the anti-sense ACR transcript are increased." In the case of MGME1 knockout in this manuscript, both 7S DNA and ACR transcripts appears to increase. Then, why this is in nice agreement with the previous results?

Lines 202-203

This sentence discusses about the results of liver mtDNA. Thus, (Fig. 2b, Supplementary Fig. 1C) should be Fig. 2c?

Lines 197-210

The authors stated that for liver mtDNA in *Mgme1* $-/-$ mice the coverage patterns are consistent with the presence of a linear deleted mtDNA fragment. On the other hand, they stated that for brain and heart mtDNA in the knockout mice, the coverage patterns are consistent with the accumulation of replication intermediates. I have concerns here. (1) Then, how are the linear deleted mtDNA fragments which were also observed in brain and heart samples explained in relation to the coverage patterns? Why do they not affect the coverage patterns in these cases? Looking at the images of Figure 2b and c, the levels of the full-length mtDNA and the partially deleted mtDNA appear to be comparable in heart, brain and liver samples from the knockout mice. (2) I understand that the authors try to infer that the replication stalling is the cause of the gradual decrease of read number (coverage) when the position of the reads goes away from O(H) in the case of liver samples. If my understanding is correct, then I wonder whether the non-specific stalling of replication can indeed be enough to influence the landscape of deep sequence of mtDNA so significantly (Figure 7d) and completely overwhelm the influence of the presence of the linear deleted mtDNA. More data would be necessary to support the authors' view to strengthen the manuscript. Otherwise, I would like to suggest that more careful interpretation and discussion of the data should be made.

In addition, to support the argument of the authors it is important to show the 2DNAGE images of mtDNA from brain samples.

In Figure 7 b,e, "bubble arc" is depicted. However, such arcs do not appear to be detected in the Southern hybridization images. Here I wish to raise concern about the intactness of the mtDNA preparations. It is an important point for this kind of study as any substantial degradation of the samples during preparation could affect the results.

Since 2DNAGE is difficult to understand for most readers, it is suggested that more explanation is added. For example, why the spot on the apex of fork arcs can be concluded as O(L)? Why the thicker arc can be interpreted to be replication stalling? Why bubble arcs are expected in O(H)-containing fragment and not in O(L)-containing fragment?

Line 261

"we report increased sequence coverage reads"
Where are the data shown in this manuscript?

Lines 273 and 274

"*Mgme1* $-/-$ mice do not accumulate point mutations (data not shown)"

Since the authors wish to say that point mutations, and not the linear deleted mtDNA, is the cause of ageing phenotype in mutator mice, demonstration of the absence of mutations in MGME1 knockout

mice is important. Thus I suggest that such data supporting the conclusion of the authors should be shown, at least in Supplementary figure.

Line 274

"do not display a progeroid phenotype"
Supporting data need to be shown.

Line 350-353

Describe the PCR condition so that others can repeat it.

Lines 356-357

Information on the probe sequences should be given. Reference 24 is cited here, but as far as I am aware the paper does not describe the details of the probes in the method section. Also, at least brief description, such as the sequence information of the primers used to generate the riboprobe is necessary for others to perform the same experiment. In addition, the description of Line 356, "For the labeling of the ACR transcrip...t" should be "For detection of the ACR transcript..."?

Line 367

"." is missing.

Supplementary Table 1

It was attached to the manuscript. However, as far as I am aware, it is not referred anywhere in the manuscript. It needs to be refereed and explained briefly.

Legends to figures need more information. I am afraid that readers may not be able to fully understand what the authors wish to tell with the figures. Comments are listed below.

Figure 1

- a. No description about colored allows.
- b. No information on the primer positions, thus impossible to follow the experiment.
- c. Comment that HSP60 is a mitochondrial protein would be helpful.

Figure 2

- a. It would be helpful if the length of PCR product with the position of primers are clearly shown with the nucleotide numbers here or somewhere in the main text. Indicate which bands are derived from normal mtDNA and deleted mtDNA. How the bands were visualized?
- b, c. Indicate which band is the full length (normal) mtDNA. Describe the details of the probe used.

Figure 3

The band patterns of mtDNA look different between panels a, b and c. Presumably treatment of the samples with restriction enzymes were performed in panel c, but not sure with panels a and b. A brief explanation of the reason would be helpful.

Figure 4

What are (-), (+) and FDR at the right of graph?

Figure 5

What is COX, ND, cytb, HSP and LSP? Non experts many not know the abbreviations. Describe the details of "NCR probe".

Figure 6

The abbreviated names of proteins and respiration complexes may not be familiar to many readers. Same applies to the expression of the supercomplexes which are added at the left of Coomassie staining. I would suggest that they should be clarified here or somewhere in the text. The explanation of L/L and L/L cre is missing.

Figure 7

"two samples" means "two independently prepared samples (derived from different mice)" (Line

603)?

As commented earlier, images of two-dimensional neutral-agarose gel electrophoresis are difficult for most readers. More information would be necessary. what are 1N and 2N?

Supplementary Figure 1

Indicate which band is the full length (normal) mtDNA. Describe the details of the probes used.

Explain the asterisks.

Supplementary Figure 2

a. Interpretation of the data is necessary. The band patterns of 3' end between the wild type and knockout also look different, but the main text (Lines 149-153) said 3' end had no changes.

b. What was cloned?

Supplementary Figure 3a

What is Bir A*? Does it mean MTS-BirA* or BirA* without MTS? Brief, but good explanation should be added.

Reviewer #2 (Expert in mtDNA mutations and mitochondrial diseases; Remarks to the Author):

The paper by Matic et al. is a very interesting and thorough study investigating the tissue-specific pattern of mtDNA replication stalling in a new mouse model of MGME1 deficiency. The authors generated a knockout mouse model to study the mtDNA maintenance defect in MGME1-related disease. Homozygous knockout mice develop mtDNA depletion and multiple deletions and the mtDNA replication stalling phenotypes were different in the tissues. Although a long linear deleted mtDNA species, similar to found in the mutator mice with POLG deficiency is present in both mouse models, the phenotype of these mice is different, since the MGME1 KO mice do not show premature aging.

I think it is an excellent and exciting study.

I have a few questions:

1. Was there any clinical phenotype observed in the MGME1 KO mice? The mtDNA maintenance defect was present in the different tissues.
2. On figure 2 the authors show result of mtDNA depletion and deletion formation in different mouse tissues, which was quite prominent in skeletal muscle and brain. Was there any muscle weakness or neurological deficit observed in these mice?
3. Was there any histological evidence of RRF/COX⁻/SDH hyperreactive fibres in muscle? Or any pathological abnormalities in brain, heart or any of the other tissues?
4. Despite the detected mtDNA depletion and deletions, the steady state level of the mitochondrial proteins remained normal in the tissues studied, which highlights that a certain threshold is needed to develop a respiratory chain defect in the cell. A comment on this would be helpful.
5. I find it very interesting that the mtDNA maintenance defect of the MGME1 KO mice is similar to the deleter mice, but the phenotype is different. The authors argue that the presence of mtDNA point mutations may be responsible for this. In which tissues and how extensively were they looking for mtDNA point mutations in the MGME1 mice?

Dear Editor,

We thank you for providing us with expert reviewers and we appreciate their constructive comments: “The paper by Matic et al. is a very interesting and thorough study... I think it is an excellent and exciting study.” (Reviewer #2).

All of the textual changes in the revised manuscript are in red font.

Our comments are as follows:

Reviewer #1 (Expert in transcription/replication; Remarks to the Author):

The manuscript entitled “Tissue-specific patterns of mtDNA replication stalling in mice lacking MGME1” by Matic et al., with Figures 1-7, Supplementary Figures 1-4 and Supplementary Table 1, describe the molecular phenotypes of MGME1 deletion in mice. They generated MGME1 knockout mice and heart and skeletal muscle-specific knockout mice and analyzed the consequences of deletion of the protein on mtDNA, 7S DNA, mitochondrial transcripts and mitochondrial proteins. The authors also analyzed the replication intermediates of mtDNA and attempted to find interacting proteins with MGME1. Although the authors performed many different experiments, I have numbers of concerns considering the standards of Nature Communications. The main concerns are described below.

(1) Similar findings regarding the effect of MGME1 deletion on mtDNA and 7S DNA were already reported in the preceding works using patient fibroblasts having MGME1 null mutation and MGME1-siRNA knockdown cells (Kornblum et al Nature Genet 2013, Nicholls et al Hum Mol Genet 2014), such as multiple deletions of mtDNA, appearance of the 11 kb sub-genomic fragment, elevated levels of 7S DNA, reduced rate of 7S DNA decay, defects in the processing of 5' termini of 7S DNA and an accumulation of replication intermediates which is similar to the results of mouse liver mtDNA in this manuscript.

Response:

Some of the data presented in the manuscript are indeed reproducing findings from previous reports using patient fibroblasts and knockdown cell lines. We feel it is important to show that our knockout model recapitulates these key features because we use it as a tool for in depth studies of pathological consequences of *Mgme1* deficiency. We would like to point out that we present many novel findings concerning the *in vivo* function of the MGME1 protein, including tissue specific replication stalling phenotypes and a new role for MGME1 protein in regulating mtDNA maintenance and expression. Moreover, the fact that our model accumulates large amounts of linear deletions without developing progeria shows that the progeria phenotype of mtDNA mutator mice is not driven by these linear deletions.

(2) The authors concluded that the long linear deleted mtDNA species is not the cause of premature ageing phenotype of mtDNA mutator mice based on the observation that MGME1 knockout mice have the linear species but do not show any ageing phenotypes. I understand that this is one of the main conclusions that this manuscript wishes to put forward strongly, as it is clearly stated in SUMMARY and DISCUSSION. However, no data regarding the physiological aspects of the MGME1 knockout mice are presented.

Response:

We now present new data showing the absence of a premature ageing phenotype in *Mgme1*^{-/-} animals. Unfortunately, the very complicated bureaucracy involved in getting ethical permits for animal studies in Germany means that it would take at least 6 months to get permission to perform additional, even simple, non-harmful phenotyping studies, such as spontaneous

locomotion in an activity box. We can only start breeding mice for this purpose once we have the ethical permit, meaning that the generation of additional phenotyping data will take at least 1 year. We feel that the new data we present (appearance, histology, fertility, blood counts – see new Fig. 3, Suppl. Fig. 3) demonstrate clearly that the *Mgme1*^{-/-} mice do not develop the severe premature ageing phenotype we see in mtDNA mutator mice. It will of course be interesting to perform more comprehensive phenotyping studies of the *Mgme1*^{-/-} mice in the future to possibly detect additional subtle aberrations.

(3) Throughout the manuscript the authors discussed about the increase/decrease of bands of DNA, RNA and proteins. However, they were not supported by the quantification data.

Response:

As suggested by the reviewer, we have now included quantification analysis for all of those experiments. (Please see new Supplementary Fig.6, new panels in the Figs. 2c, 5b and Supplementary Fig.1e).

(4) The authors observed difference in the replication intermediates of mtDNA from two organs, liver and heart of the knockout mice using 2DNAGE. The difference is interesting. A strong spot at O(L) was observed in the liver mtDNA, but not in the heart mtDNA, of the mutant mice. However, why such differences are caused was not addressed.

Response:

The observed tissue-specificity of our *Mgme1*^{-/-} mouse model is in agreement with the well-known tissue heterogeneity of mitochondrial disorders that is largely unexplained and our mouse model can be a useful tool to study this phenomenon *in vivo*. In the discussion section (lines 340-348) we hypothesize that differential regulation of dNTP metabolism might contribute to those tissue-specific phenotypes and to test this we assessed steady state levels of various mitochondrial dNTP homeostasis proteins in heart and liver tissue of our model. As documented in the figure below, we find no expression differences of mitochondrial dNTP homeostasis proteins between *Mgme1*^{-/-} and control mice in different tissues. The analyses involved assessment of the protein expression of ENT1 (mitochondrial nucleoside uptake), TK2 (kinase of pyrimidine branch of mitochondrial salvage pathway), AK2 and DGUOK (mitochondrial purine salvage pathway) and SUCLA2 (ATP-dependent succinate-CoA ligase) in both heart and liver.

(5) In SUMMARY, the authors stated, "we also report a role for MGME1 in the regulation of replication and transcription termination at the end of the control region of mtDNA". I am afraid it is an overstatement. They analyzed 7S DNA, transcripts in the non-coding region of mtDNA and mitochondrial transcripts. Then, from the data they speculated the possible role of MGME1, but not demonstrated it.

Response:

We do agree with the reviewer that more data are needed to clarify the role of MGME1 at the end of the mitochondrial control region. However, we feel that several lines of evidence pointing towards a role for MGME1 in regulation of mtDNA replication and H-strand

transcription constitute important new information that will provide a basis for additional mechanistic studies in the future.

(6) I am afraid that it is difficult to follow and understand the content of the manuscript as the description (explanation) of the data are not sufficient in the text and legends to figures.

Response:

We have included more details in the figure legends and text as suggested by the reviewer.

(7) The whole set of data that were obtained from the knockout mice would be helpful for other researchers who are interested in MGME1 proteins. However, since the protein has been already studied both in vitro and in living cells extensively, the impact of this work to the relevant field is not strong enough to be considered in Nature Communications.

Response:

We respectfully disagree with the reviewer as our knockout model provides a number of new interesting findings of importance for the understanding of MGME1 function. Furthermore, the mouse model we describe here will constitute a novel powerful tool for future studies.

Overall, I am afraid but this manuscript does not meet the standards of Nature Communications. I attached my specific comments below with a hope of helping the authors to prepare a new manuscript elsewhere.

Specific comments

Lines 60-63

I am afraid that the content of the sentence is ambiguous. At least a couple of 'extensive in vitro work' should be described to help readers' understanding. Furthermore, a number of animal models, not only mouse but also other species were already generated and intensively studied in the past and many important insights were gained. Thus, it would be helpful if such preceding animal studies are introduced briefly here as a comparison of the current work.

Response:

We followed suggestions of the reviewer and quoted and described importance of some of the previous in vitro and in vivo work studying mtDNA maintenance disorders. (Lines 61-65).

Line 70

Is the reference correct?

Response:

We have corrected this reference.

Lines 92-98

"homozygous knockout (Mgme1^{-/-}) mice that appeared healthy when followed until the age of 12 months."

"... heart- and skeletal-muscle-specific Mgme1 knockout mice (Mgme1^{loxP/loxP}; +/-Ckmm-cre) that had no obvious phenotype when followed until 12 months of age."

One of the conclusions which this manuscript put forward strongly is that knockout of MGME1 does not cause any ageing phenotype. However, no data related to it is shown. If the authors wish to make such a proposition, data that support the statements should be sufficiently presented. As the authors mentioned "animal models are nevertheless essential to understand the in vivo metabolic consequences" in Introduction, the strong point of animal model studies is that they enable researchers to study physiological and behavioral aspects of the gene of interest using the animals. Therefore, such data are expected to be presented in this manuscript.

Furthermore, discussion regarding the similarity/difference of the phenotypes of the mice and patients with *Mgme1* mutations is lacking. I believe it is another strong point of animal model studies and thus such discussion should be made.

Response:

Numerous studies suggest that mtDNA mutations and deletions contribute to the ageing phenotypes in experimental animals and humans. Despite the accumulation of an 11kb linear mtDNA fragment spanning the entire major arc of the mtDNA (previously described in the prematurely ageing mtDNA mutator mouse), the MGME1 knockout mice show no premature ageing phenotypes. At the age of 40 weeks (see Fig. 2 in Trifunovic et al Nature 2004:429:417-423) mutator mice show very marked premature ageing. In contrast, the MGME1 knockout mice look strikingly normal at this age, see photos below.

Moreover, all of the mtDNA mutator mice are dead before the age of 52 weeks, whereas the oldest mice *Mgme1* mice we currently have are 70 weeks old and show no obvious signs of premature ageing, see photos below.

The prematurely ageing male mtDNA mutator mice are infertile with smaller testes, reduced sperm content and reduced sperm motility (Trifunovic et al Nature 2004:429:417-423; Jiang et al Cell Met. 2017:26:429-436). In contrast, MGME1 knockout mice are fertile with normal testis morphology, normal testis weight, normal sperm count and normal sperm motility (new Fig. 3).

At ~11 months of age, the mtDNA mutator mice suffer from severe anaemia with reticulocytosis and blood haemoglobin concentration as low as 50 g/l⁻¹. In contrast, MGME1 knockout mice have normal haemoglobin concentration at 20 weeks of age and a slight decrease without reticulocytosis at 70 weeks of age (new Supplementary Fig. 3).

Finally, we also include new results showing histological analysis of various tissues from 20 and 70 week old MGME1 knockout mice (new Fig. 3, Supplementary Fig.3).

Lines 77-78

“MGME1 is not essential for mouse embryonic development”

In relation to above, data supporting the statement should be presented, such as no significant

difference in the birth rate of +/+ and -/- ?

Response:

This information is now provided it in the revised manuscript (lines 94-96).

Line 95 and Figure 1c

In Figure 1c, how the band indicated by a line was confirmed to be MGME1? Convincing data should be presented to demonstrate the identity of the band. This is important as another band which is migrated slightly faster is also clearly detected by the antibody against MGME1. For example, siRNA (more than one siRNA should be used) knockdown of MGME1 gene in cultured mouse cells would be a good and straightforward experiment to confirm the identity of the band.

Evidence of successful knockout in other organs than heart in the -/- mice needs to be shown by similar western blotting. Sufficient data to demonstrate the successful tissue-specific knockout in heart and skeletal muscle specific knockout mice should be also shown.

Response:

We now provide new data showing that the MGME1 protein is missing in various tissues of the *Mgme1*^{-/-} mice as well as in the hearts of the tissue-specific knockouts (Fig. 1c). As pointed out by the reviewer, the antibody against MGME1 also detects slightly faster migrating band. However, this band is likely due to tissue-specific cross-reaction because it is present in most *Mgme1*^{-/-} tissues lacking the MGME1 protein (Fig. 1c). The MGME1 protein has an expected molecular weight of 36kDa and this is in good agreement with the apparent size of MGME1 on western blots and the migration of the slightly larger recombinant tagged MGME1 protein of 39kDa (see Figure below).

Lines 101-102

Readers will find it difficult to understand how the gel image of Fig. 2a demonstrate multiple deletion of mtDNA in -/- samples as it is not explained. Proper explanation of the data should be described in the text or the legend to the figure.

Response:

We have now included detailed explanation of long-extension PCR in the results section (lanes 107-110) and in the figure legend.

Line 104

“showed both mtDNA depletion... (Fig. 2b,c)”

To confirm the authors' conclusion, quantification of the band should be shown as graphs with sufficient number of samples with statistical analysis. Provision of quantification data with statistical analysis should apply to all other data sets with which the authors wish to discuss about increase/decrease and make quantitative interpretation/discussion.

In addition, quantification of the comparison of the band intensity of the intact mtDNA and

~11 kb in MGME1 KO samples (and polg mut/mut samples) would be informative when readers consider similarity/difference between MGME1 KO mice and polg mut/mut mice.

Response:

We have followed the suggestion by the reviewer and have now quantified mtDNA levels in wildtype and *Mgme1*^{-/-} mice (new panel Fig. 2c). The *Mgme1*^{-/-} mice exhibit marked depletion of mtDNA. We now also provide quantification with statistical analyses of other data sets (new panel Fig. 5b and new Supplementary Figs.2 and 6). In addition, we present quantification of levels of linear deleted mtDNA on Southern blots showing that the 11kb deletion is more abundant in *Mgme1*^{-/-} than in *Polg*^{mut/mut} mice (Supplementary Fig. 1e).

Fig. 2b,c and S. Fig. 1b, c

How the sizes of the bands were identified so accurately (7.1 and 3.8 and 8.4 and 2.6 kb indicated at the left side of Southern blotting images in Fig. 2 and S Fig. 1b, respectively)? It would be informative if such information is added; for example, DNA ladder (size marker) that was electrophoresed with the samples on the same gels should be shown.

Response:

The approximate sizes of the bands are identified by calculation of the fragment size generated after the restriction digestion of the linear deletion and comparison with a size marker (see figure bellow). We prefer not to include molecular size markers in the figure mostly because of space issues.

In lane 6 in Fig. 2b, the molecular sizes of the two bands with asterisks appear to be different from those in lanes 2 and 4. Are they the same species?

Response:

We believe that those are same species but the slightly faster migration in the gel is the consequence of the somewhat higher DNA amount loaded in those lanes.

S. Fig. 1c

The lower portion of the gel image needs to be shown in the similar way to S. Fig. 1b to demonstrate the absence of any smaller bands in -/- samples.

Response:

We have corrected this figure by including larger area of the gel in the figure (see new Fig. 1c).

Lines 115-116

“The linear deleted mtDNA molecules are very similar in size”

No appreciable bands were seen in mut/mut PolgA sample lanes in Fig. 2b and S. Fig. 1b.

Longer exposure images should be shown.

Response:

We have included longer exposure of the images (Fig. 2b and Supplementary Fig. 1b) and also included new panels in Supplementary Fig. 1d where the linear deletion in samples from the mtDNA mutator mice is more obvious as the Southern blot analysis was performed using pure mitochondrial DNA.

Lines 127-128

“By using Southern blot analyses, we found increased steady-state levels of 7S DNA in *Mgme1*^{-/-} mice (Fig. 3a).”

I am afraid that in Fig. 3a 7S DNA itself does not appear to increase significantly. The apparent increase of the band intensity of 7S DNA may be an effect of the stronger background smear in ^{-/-} lanes. To eliminate the ambiguity, it is suggested that a DNA probe which does not give such smear is used and then the quantification data are provided. Or, sufficient explanation should be provided how the authors came to their conclusion with the current image.

Response:

We agree with the reviewer that signal for 7S DNA in *Mgme1*^{-/-} samples is not as focused as in wildtype controls. However, we always see this kind of signal in *Mgme1*^{-/-} samples and it is likely the consequence of the presence 7S DNA molecules of various sizes. We now provide a quantification (n=4) showing a significant increase of 7S DNA in *Mgme1*^{-/-} samples (new panel Fig. 5b).

Lines 156-170

The BioID would be a powerful tool to identify proteins that interact with the protein (in this case MGME1) fused with biotin ligase in living cells. However, since the fusion protein is expressed exogenously (and presumably overexpressed), it may be possible that some proteins that are not interacted with MGME1 under physiological conditions are captured under BioID conditions. Thus, if authors wish to put forward the conclusion that POLRMT, TWINKLE, SSPB1 and POLGA interact with MGME1, it is required that the interactions should be supported with another experiment, such as CoIP. Otherwise, the authors should rewrite the paragraph carefully with consideration of how far one can say with BioID only. This applies to all the sentences which is related to the data elsewhere in the manuscript.

Response:

The BioID approach is a standard procedure that allows the identification of weak and/or transient protein-protein interactions. Such interactions are likely present among the key enzymes involved in mtDNA replication. In support of the validity of the BioID approach, we would like to point out that the MGME1- POLGA also has been identified by expression of MGME1-FLAG in HEK293T cells (Nicholls et al NAR 2014:23:6147-6162). Furthermore, with unpublished CoIP experiments we could confirm the MGME1- POLGA interaction. However, we were not able to identify additional interacting partners, likely due to their transient nature.

Lines 172-183.

It is necessary to show the quantification graphs of all RNAs which are discussed in the paragraph.

Response:

Quantification of all the RNAs is shown in the new Supplementary Fig.6.

In Figure 5c, the lower band is indicated as 7S RNA. Then what is the upper band?

Explanation and/or supporting data on how the authors can know the indicated band is 7S RNA would be necessary.

Response:

The presence of double band on the northern blot is likely a consequence of incomplete denaturation of the RNA in this particular experiment. We have repeated experiment after complete denaturation of the RNA and have exchanged the panel (new Fig. 7c).

It would be helpful if ACR transcript is indicated clearly in Figure 5a. Here, most readers would not be able to follow the authors' argument as ACR transcript is not sufficiently explained.

Response:

We have indicated ACR transcript using dashed boxes (new Fig. 7d). We do agree with the reviewer that ACR transcript was poorly described in the result section and we have clarified this in more detail in the revised manuscript (lines 216-218).

Lines 182-183.

“This finding is in nice agreement with previous results that in the absence of premature mtDNA replication termination (i.e. 7S DNA) the levels of the anti-sense ACR transcript are increased.”

In the case of MGME1 knockout in this manuscript, both 7S DNA and ACR transcripts appears to increase. Then, why this is in nice agreement with the previous results?

Response:

The steady state levels of 7S DNA are indeed increased in *Mgme1*^{-/-} mice (due to their increased stability as documented by ddC treatment experiment presented in Fig.5d). However, as shown by the *in organello* replication experiment (Fig. 5c) the *de novo* synthesis of 7S DNA is severely impaired in mitochondria lacking MGME1. Both synthesis of 7S DNA and termination of H-strand transcription are severely impaired in *Mgme1*^{-/-} mice pointing out a role for MGME1 at the end of the control region of mtDNA.

Lines 202-203

This sentence discusses about the results of liver mtDNA. Thus, (Fig. 2b, Supplementary Fig. 1C) should be Fig. 2c?

Response:

We have corrected this error in the revised version of the manuscript.

Lines 197-210

The authors stated that for liver mtDNA in *Mgme1* ^{-/-} mice the coverage patterns are consistent with the presence of a linear deleted mtDNA fragment. On the other hand, they stated that for brain and heart mtDNA in the knockout mice, the coverage patterns are consistent with the accumulation of replication intermediates. I have concerns here. (1) Then, how are the linear deleted mtDNA fragments which were also observed in brain and heart samples explained in relation to the coverage patterns? Why do they not affect the coverage patterns in these cases? Looking at the images of Figure 2b and c, the levels of the full-length mtDNA and the partially deleted mtDNA appear to be comparable in heart, brain and liver samples from the knockout mice. (2) I understand that the authors try to infer that the

replication stalling is the cause of the gradual decrease of read number (coverage) when the position of the reads goes away from O_H in the case of liver samples. If my understanding is correct, then I wonder whether the non-specific stalling of replication can indeed be enough to influence the landscape of deep sequence of mtDNA so significantly (Figure 7d) and completely overwhelm the influence of the presence of the linear deleted mtDNA. More data would be necessary to support the authors' view to strengthen the manuscript. Otherwise, I would like to suggest that more careful interpretation and discussion of the data should be made.

Response:

It is important to note that the coverage curves cannot be used as a perfectly quantitative measure of the amount of the linear deletion. During library preparation, the linear fragments are likely more efficiently incorporated into the sequencing libraries. The data we have published previously support such a bias because the mtDNA mutator mouse show a similar coverage curve where the relative coverage elevates from 0.6 to 0.9 in the region containing the linear deletion, corresponding to ~30% deleted mtDNA (Ameur et al. PLoS Genet, 11:e1005333). However, we show in our manuscript that the mtDNA mutator mouse has only ~20% of the linear deletion on Southern blots (Supplementary Fig. 1 d and e). Given this discrepancy, we performed 2D gel electrophoresis to more accurately assess the relative amounts of stalled replication intermediates. Furthermore, we would like to point out that the 11 kb linear deletion and the nonspecific replication stalling both produce an increase in read coverage across the major arc. The replication stalling produces a more gradual effect than the linear deletion.

In addition, to support the argument of the authors it is important to show the 2DNAGE images of mtDNA from brain samples.

Response:

We have now included 2DNAGE gels performed on brain samples (new panels in the Supplementary Fig. 7 b and c). In line with the sequence coverage patterns (Fig.9d, Supplementary Fig. 7a) we see similar replication stalling profiles in brain and heart samples.

In Figure 7 b,e, "bubble arc" is depicted. However, such arcs do not appear to be detected in the Southern hybridization images. Here I wish to raise concern about the intactness of the mtDNA preparations. It is an important point for this kind of study as any substantial degradation of the samples during preparation could affect the results.

Response:

It is true that we would have expected to have seen a more prominent bubble arc in the O_H containing fragments. Replication bubbles are very fragile and susceptible to nicking during preparation from solid tissues and subsequent treatments. In this case nicked bubbles run within the fork arc. This does not, however, alter the conclusions of the experiment.

Since 2DNAGE is difficult to understand for most readers, it is suggested that more explanation is added. For example, why the spot on the apex of fork arcs can be concluded as O_L ?

Response:

The apex of the fork arc represents a replication intermediate in which the replication fork is in the middle of the restriction fragment. In the blots presented in Fig. 9, the prominent spot on the fork arc lies slightly before the apex, corresponding to a point slightly before the middle of the fragment (with the direction of replication coming around the major arc). Within the resolution possible using the technique, the location of O_L (approx. nt. 5,200) matches well with this position (around 1.9 kb from the BclI site at nt. 7,084 and around 2.1 kb from the site at nt. 3,102). The location of O_L in this digest can also be compared with that

in the mtDNA mutator mouse, which also shows stalling at this point (Figure 5 of Bailey et al. (2009) *Nucleic Acids Res.*; 37(7): 2327-35), although over a broader region compared with the MGME1 knockout.

Why the thicker arc can be interpreted to be replication stalling?

Response:

Each point on the fork arc corresponds to a point within the restriction fragment. The increase in signal intensity at all points on the fork arc corresponds to more replication intermediates at all points along the fragment. As the copy number does not increase in the knockouts, this increased density of replication forks can be interpreted as replication proceeding slower, or stalling in a non sequence specific manner.

Why bubble arcs are expected in O(H)-containing fragment and not in O(L)-containing fragment?

Response:

Bubble arcs are produced from strand separation and replication initiation within the restriction fragment. When a replication fork leaves the restriction fragment then the intermediates become forks and run within the fork arc. Replication initiation from O_L does not involve bubble formation, as an advancing replication fork from the major arc must expose the O_L region before DNA synthesis can begin. We have modified the results section related to 2DNAGE to give a bit more explanation for the phenotypes (lines 246-253).

Line 261

“we report increased sequence coverage reads” Where are the data shown in this manuscript?

Response:

This sentence refers to the Fig 9a and d as well as Supplementary Fig. 7a. The raw data will be deposited into a ENA sequencing repository.

Lines 273 and 274

“Mgme1^{-/-} mice do not accumulate point mutations (data not shown)”

Since the authors wish to say that point mutations, and not the linear deleted mtDNA, is the cause of ageing phenotype in mutator mice, demonstration of the absence of mutations in MGME1 knockout mice is important. Thus I suggest that such data supporting the conclusion of the authors should be shown, at least in Supplementary figure.

Response:

This is important point raised by both reviewers. We have therefore measured the levels of point mutations in various tissues of *Mgme1^{-/-}* mice at 70 weeks of age as well as in liver at 11 weeks of age. The new data we present confirm that there is no increase of levels of mtDNA point mutations in skeletal muscle or highly proliferative tissues, such as spleen and liver (Supplementary Fig.2 a and b).

Line 274

“do not display a progeroid phenotype”

Supporting data need to be shown.

Response:

Please see response above (response to “Lines 92-98”).

Line 350-353

Describe the PCR condition so that others can repeat it.

Response:

We have included protocol for performing this PCR in the methods section.

Lines 356-357

Information on the probe sequences should be given. Reference 24 is cited here, but as far as I am aware the paper does not describe the details of the probes in the method section. Also, at least brief description, such as the sequence information of the primers used to generate the riboprobe is necessary for others to perform the same experiment.

Response:

We have included Supplementary table containing all this information (Supplementary table 2).

In addition, the description of Line 356, "For the labeling of the ACR transcript...t" should be "For detection of the ACR transcript...?"

Response:

We have corrected this sentence in the manuscript.

Line 367

“.” is missing.

Response:

We have corrected this typo in the manuscript.

Supplementary Table 1

It was attached to the manuscript. However, as far as I am aware, it is not referred anywhere in the manuscript. It needs to be refereed and explained briefly.

Response:

We now refer to this table in the revised manuscript.

Legends to figures need more information. I am afraid that readers may not be able to fully understand what the authors wish to tell with the figures. Comments are listed below.

Figure 1

a. No description about colored allows.

Response:

We have included description of the colours.

b. No information on the primer positions, thus impossible to follow the experiment.

Response:

We have included schematic representation of *Mgme1* cDNA and indicated position of the primers (Fig. 1b).

c. Comment that HSP60 is a mitochondrial protein would be helpful.

Response:

We have included this information.

Figure 2

a. It would be helpful if the length of PCR product with the position of primers are clearly shown with the nucleotide numbers here or somewhere in the main text. Indicate which bands are derived from normal mtDNA and deleted mtDNA. How the bands were visualized?

Response:

We included requested information.

b, c. Indicate which band is the full length (normal) mtDNA. Describe the details of the probe used.

Response:

We have included the requested information in the figure legend. The probe details are listed in Supplementary table 2.

Figure 3

The band patterns of mtDNA look different between panels a, b and c. Presumably treatment of the samples with restriction enzymes were performed in panel c, but not sure with panels a and b. A brief explanation of the reason would be helpful.

Response:

We have provided this information. Indeed, the difference is coming from the fact that DNA in panels a and c (new panels a and d) is digested by restriction enzyme and the DNA in panel b (new panel c) is non-digested (new Figure 5).

Figure 4

What are (-), (+) and FDR at the right of graph?

Response:

We have provided this information in the figure legend.

Figure 5

What is COX, ND, cytb, HSP and LSP? Non experts many not know the abbreviations. Describe the details of "NCR probe".

Response:

We have provided this information in the figure legend and Supplementary table 2.

Figure 6

The abbreviated names of proteins and respiration complexes may not be familiar to many readers. Same applies to the expression of the supercomplexes which are added at the left of Coomassie staining. I would suggest that they should be clarified here or somewhere in the text. The explanation of L/L and L/L cre is missing.

Response:

We have provided this information in the Supplementary table 2.

Figure 7

"two samples" means "two independently prepared samples (derived from different mice)" (Line 603)?

Response:

We have provided this information in the figure legend.

As commented earlier, images of two-dimensional neutral-agarose gel electrophoresis are difficult for most readers. More information would be necessary. what are 1N and 2N?

Response:

We have provided this information in the figure legend.

Supplementary Figure 1

Indicate which band is the full length (normal) mtDNA. Describe the details of the probes used. Explain the asterisks.

Response:

We have provided the requested information in the figure legend.

Supplementary Figure 2

a. Interpretation of the data is necessary. The band patterns of 3'end between the wild type

and knockout also look different, but the main text (Lines 149-153) said 3' end had no changes.

Response:

The reviewer is correct. The 3' ends were not identical between *Mgme1*^{-/-} mice and controls, but the major changes occurred at the 5' ends, as previously reported (Nicholls, T.J., et al. Hum. Mol. Genet. 2014, 23: 6147–6162. 2014). Therefore we have modified the text to emphasize this fact (lines 183-186).

b. What was cloned?

Response:

We used purified mtDNA to capture the 5' and 3' ends of the 7S DNA onto which we ligated adaptor oligos using T4 RNA ligase to enable specific capture of ssDNA ends but not dsDNA (Troutt, A.B., et al. Proc. Natl. Acad. Sci. USA. 89. 9823-9825. 1992). The captured DNA fragments were amplified by PCR using one primer designed to bind the ligated adaptor and another designed to bind the flanking mtDNA. The amplified DNA was blunt-ended and cloned directly into the pJET1.2/blunt plasmid. These methods are described in the Supplemental Experimental Procedures.

Supplementary Figure 3a

What is Bir A*? Does it mean MTS-BirA* or BirA* without MTS? Brief, but good explanation should be added.

Response:

BirA* stands for promiscuous biotin ligase without mitochondrial targeting sequence. We explained this in the figure legend.

Reviewer #2 (Expert in mtDNA mutations and mitochondrial diseases; Remarks to the Author):

The paper by Matic et al. is a very interesting and thorough study investigating the tissue-specific pattern of mtDNA replication stalling in a new mouse model of MGME1 deficiency. The authors generated a knockout mouse model to study the mtDNA maintenance defect in MGME1-related disease. Homozygous knockout mice develop mtDNA depletion and multiple deletions and the mtDNA replication stalling phenotypes were different in the tissues. Although a long linear deleted mtDNA species, similar to found in the mutator mice with POLG deficiency is present in both mouse models, the phenotype of these mice is different, since the MGME1 KO mice do not show premature aging.

I think it is an excellent and exciting study.

We appreciate reviewer's positive comments on our study.

I have a few questions:

1. Was there any clinical phenotype observed in the MGME1 KO mice? The mtDNA maintenance defect was present in the different tissues.

Response:

The young *Mgme1*^{-/-} mice do not display any obvious clinical phenotype despite a clear phenotype on the molecular level. The patients carrying *MGME1* mutations are mostly adults, however initial clinical symptoms were seen also in childhood (Kornblum et al, Nat. Genet. 2013, 45:214-219). Recently, a child with early onset cerebellar ataxia and a novel frameshift deletion in *MGME1* was described (Hebbar et al, 2017, Eur J Med Genet 60:533-535).

This discrepancy between phenotypes in humans and mice has been reported also in other studies, e.g. *SURF1* gene mutations cause a severe COX deficiency and Leigh syndrome in humans, whereas in *SURF1*^{-/-} knockout mice only have a mild COX defect (Dell'agnello C et al 2007 Hum. Mol. Genet. 16:431-44.).

To further analyse phenotypes in *Mgme1*^{-/-} mice, we have performed a number of additional experiments as outlined above in the comments to reviewer 1.

2. On figure 2 the authors show result of mtDNA depletion and deletion formation in different mouse tissues, which was quite prominent in skeletal muscle and brain. Was there any muscle weakness or neurological deficit observed in these mice?

Response:

As pointed out above, administrative issues concerning ethical permits prevent us from performing more detailed phenotyping of *Mgme1*^{-/-} mice within a reasonable time frame.

3. Was there any histological evidence of RRF/COX -/SDH hyperreactive fibres in muscle? Or any pathological abnormalities in brain, heart or any of the other tissues?

Response:

We found COX negative cells in heart and colon samples of old *Mgme1*^{-/-} mice (new Fig.4). We also analysed skeletal muscle samples but could not find any COX negative fibers. The results from brain were inconclusive because of technical issues and have to be repeated in the future after ageing of new cohorts of mice.

4. Despite the detected mtDNA depletion and deletions, the steady state level of the mitochondrial proteins remained normal in the tissues studied, which highlights that a certain threshold is needed to develop a respiratory chain defect in the cell. A comment on this would be helpful.

Response:

We fully agree with the reviewer that this is a very important point and we have commented on this in the revised manuscript version (lanes 233-236).

5. I find it very interesting that the mtDNA maintenance defect of the MGME1 KO mice is similar to the deleter mice, but the phenotype is different. The authors argue that the presence of mtDNA point mutations may be responsible for this. In which tissues and how extensively were they looking for mtDNA point mutations in the MGME1 mice?

Response:

This point was raised by both reviewers and we have measured the levels of point mutations in liver, spleen and skeletal muscle tissues (please see response to reviewer 1- response to Lines 273 and 274) and (Supplementary Fig.2 a and b).

Reviewers' comments:

Reviewer #1 (Remarks to the Author):

The revised manuscript entitled "Tissue-specific patterns of mtDNA replication stalling in mice lacking MGME1" by Matic et al. responded many of the concerns I raised. However, I still have a number of concerns on the technical aspects of data and the interpretation/description of them. They are listed below.

Major point 1. Data of 2DNAGE

If replication bubble arcs in the samples subjected to 2DNAGE are completely degraded (Fig. 9b,e and S.Fig. 7b), there would be no guarantee that replication fork arcs used for the ground of replication stalling in some, but not all, tissues of -/- animals did not suffer any damage in some sample preparations but not every preparation. I respectfully disagree with the authors' comment in their rebuttal letter that the degradation of bubble arcs does not alter their conclusion if they do not have any supporting experimental evidence of the claim. The authors also commented that replication bubble arcs are very fragile. There are, however, examples that detected bubble arcs, fork arcs and other types of replication arcs from solid tissue mtDNA preparation. For instance, Bailey et al NAR 2009: 37: 2327-35 showed bubble arcs from Mutator mice. Other examples are seen in Pohjoismäki et al JMB 2010: 397: 1144-55, Pohjoismäki et al JBC 2009: 284: 21446-57, Yasukawa et al EMBOJ 2006: 25: 5358-71 and so forth. Therefore, I believe that it is much more convincing if the difference in replication is discussed with 2DNAGE that were performed with intact mtDNA preparations or at least preparations with little sign of major degradation (i.e., complete absence of replication bubbles). Another concern on the comparison of the intensity of the fork arcs is that no information was provided on whether the same amount of mtDNA were loaded to the gels between samples. For example, in Fig. 9c, the 1N spot in +/+ appears to be stronger than that in -/-. (From Method section I am aware that 3 ug of mtDNA was subjected to restriction enzyme digestion. However, recovery efficiency of DNA from precipitation step after the enzyme step is not always 100%). Thus, when one discusses about replication stalling based on the intensity of the arcs and spots on the arcs, a convincing way would be that the intensities of arcs/spots are standardized/normalized by those of 1N spots (I understand it is derived from non-replicating fragments) and the values of the arcs (spots)/1N are compared between +/+ and -/-. The manuscript does not describe whether the authors compared the fork arcs with such quantitative analysis. Or, the amount of mtDNA loaded to the gels should be precisely adjusted and the arcs and spots on the arcs are quantified (without 1N spot standardization, if appropriate) and compared. Also, no information is provided whether the series of 2DNAGE experiments were performed repeatedly to confirm biological reproducibility.

Therefore, I would suggest the followings to confirm the technical soundness of 2DNAGE analysis.

- (1) Produce higher quality 2DNAGE images with biological n=3 or more. Then, perform quantifications and present the results of fork arcs/1N and OL spot/1N. Or, confirm that the same amount of mtDNA preparation was run between +/+ and -/- and quantify the intensity of the fork arcs and OL spots. For example, a same portion of fork arc between panels can be quantified for the fork arc quantification. Then present them as graphs.
- (2) Or, perform above-described quantitative analysis to the existing 2DNAGE images (with biological triplicates or more).
- (3) Or, for some reason if any of above quantification is not possible, at least it is necessary to present the data as Supplementary information that demonstrate fully the reproducibility of the results of Fig 9b,c,e,f and S.Fig. 7b,c (with biological triplicates or more).

Major point 2. Interpretation of brain mtDNA 2DNAGE data in the text does not appear to be supported by the data.

The newly provided 2DNAGE image of brain -/- sample (S.Fig 7c). has a spot on the fork arc that

appears to be considerably stronger than the one on heart mtDNA (Fig.9f). The spot on S.Fig. 7c panel looks to me more similar intensity to that on Fig. 9c than that on Fig. 9f. However, this is not appreciated in the text but the gel image of brain mtDNA (S.Fig. 7c) was, like the heart case, interpreted to be consistent with the gradual decrease of read number from OH to OL. I feel that the data do not support the interpretation. To me, the brain mtDNA results suggest that the occurrence of the strong pause at OL is not related to the coverage patterns but rather suggests the presence of complex nature of replication defect variation and coverage distribution variation between tissues, which details remain unknown.

Major point 3. Re: response to my major concern (2) in my original comment

Figure presentation of photographs of the animals would be necessary to support the statement "Mgme1 $-/-$ mice had a normal gross appearance (Line 96)". I believe that such photos are available as some of them are nicely shown in the rebuttal letter.

Major point 4. Identity of western blotting band.

It relates to Fig. 1c, and to my original comment to Line 95 in the original manuscript. I am afraid that the new data provided in the revised Fig. 1 does not constitute the evidence that the band indicated by the authors is MGME1 protein. Independent evidence that the indicated band is MGME1 enables the authors to confirm the successful knockout of MGME1 in their $-/-$ animals at the protein level, but disappearance of the indicated band in the $-/-$ animal itself cannot be used as the evidence that the band is MGME1. That was why I suggested siRNA knockdown of MEMG1, an independent experiment to show which band is the protein, in my original comments. I believe it is an important control experiment since the slightly faster migrating band (indicated as an asterisk) also disappears in spleen, liver and kidney of $-/-$ animals and decreases in the heart. To my eyes, the faint bands in $-/-$ of liver and kidney (the new Fig. 1c) and those in $-/-$ of heart (the old Fig. 1c) migrated at slightly different positions than the asterisk band in $+/+$ counterparts. I am aware that the faster migrating band were run at the same position in $+/+$ and $-/-$ without reduction in the $-/-$ band intensity in the image in the rebuttal letter. However, images in the old and new Fig. 1c give different impression to me. What I fear is a possibility that both bands are actually MGME1 (i.e., splice variants). This point will not overturn the whole manuscript, but since MGME1 is the target protein of this work, I believe that clear identification of MGME1 band would greatly support the work.

Major point 5. Re: response to my major concern (3)

Since most of quantification data are provided after revision, the manuscript reads better now. However, the presentation is not sufficiently clear as important information for understanding the figures is missing (explained below).

Fig. 2c: Does mtDNA signal (Line 647) mean "full length mtDNA signal"? Since mtDNA from $-/-$ animals show prominent sub-16kb fragments in Fig. 2b, it is informative to clearly describe this point. Also, describe the details of the data used to produce the graph. Describe whether $n=3$ is biological repeats or not.

S.Fig. 2a and b: The number (n) of the animals analyzed is missing. Also, describe whether n is biological repeats or not. I presume that the data from liver mtDNA of 70w Mgme1 $+/+$ and 70w Mgme1 $-/-$ in S.Fig.2a and S.Fig.2b were independently obtained from different mice as they are shown as the separate panels. State this point clearly.

Fig. 3: Describe the details of whether $n=3$ is biological repeats or not. Also, describe whether " $n=3$ " applies to all the panes a-d. (Does it also apply to a as well?)

Fig. 4: Three triplicates (Line 661) means three sections from heart, colon and S-muscle from three different animals [= (technical triplicates) x (biological triplicates)]? Describe this point clearly. Also add the information of the week age of Mgme1 $+/+$.

Fig. 5c: Describe which images were quantified. Also, describe whether $n=4$ is biological repeats or not. Provide the quantification data that support the sentence of Lines 173-174.

Fig. 5d: Provide a graph for the experiment [7SDNA/mtDNA ratio (and mtDNA amount, if necessary) of – and + ddC of Mgme1+/+ and Mgme1-/-] to quantitatively demonstrate slower rate of decline of 7S DNA against mtDNA in -/-. This would be necessary to support the authors' statement in the second paragraph of Page 7.

S.Fig. 3a-c: The number (n) of the animals analyzed is missing. Also, state whether n is biological repeats or not.

S.Fig. 6 Describe the details of whether n=3 is biological repeats or not.

Major point 6. Newly provided long exposure images of polgA mut/mut samples do not clearly show the additional bands

It relates to Line 123-125, in relation to my original comment to Lines 115-116 of the original manuscript. A long exposure image is now provided in Fig. 2. However, the lower additional bands with an asterisk specific to polgA mut/mut is very difficult to see. At least to me, there is no band at the exact position of lower asterisk. To support the authors' statement of this sentence, presentation of stronger images on which the two bands with asterisks can be seen easily is necessary. Similarly, it is not possible to discern a band at the position of the asterisk in S.Fig. 1. In this case, it appears that better separation of the bands is required with lower percentage gel electrophoresis, or at least please show an image that one can appreciate the presence of the band.

Major point 7. The statement on Line 143, "Importantly, the anemia of Mgme1-/- mice is much milder than the anemia in mtDNA mutator mice" is not supported by any data.

Please show the data, or indicate a reference paper showing the data, to support this important statement. I expect that the authors can present them without a new animal experiment as the rebuttal letter says, "the mtDNA mutator mice suffer from severe anaemia with reticulocytosis and blood haemoglobin concentration as low as 50 g/l."

Major point 8. The conclusive statement, "we found increased steady-state levels of 7S DNA in Mgme1-/- mice (Fig. 5a, b)." on Lines 160-161 is not satisfactory supported by the data provided.

The authors replied to my comment (comment to Lines 127-128 of the original manuscript), however I am afraid that their response does not clear my concern. The authors commented the possible reason in their response, but it is not supported by any data. I suggested the use of other probes to obtain clearer images for quantification, but whether it was attempted is not known.

Since the southern blot images of Fig. 2 and S.Fig. 1b,c show weak background smear at low molecular weight region of the gels even in -/- samples compared to Fig. 5a, the authors should be able to obtain the information of 7S DNA from the longer-exposure images of Fig. 2 and S.Fig. 1b,c, if not with a different hybridization probe. In any case, I consider that the accurate quantification of 7S DNA is important because better images might give a different, if not completely different, graph (and therefore interpretation) on 7S DNA steady state levels in the case of Mgme1 knockout mice. Or, the sentence should be rewritten to explain the data carefully (as I suggested in my original comment). For example, "although the smear around 7S DNA in Mgme1-/- mice samples prevented us from accurate quantification of 7S DNA band, southern blot analyses suggested an increase of steady-state levels of 7S DNA against full-length mtDNA in Mgme1-/- mice."

Major point 9. The statement regarding BioID

It relates to Lines 190-204, and to my original comment to Lines 156-170 of the original manuscript. The authors stated in their rebuttal letter, "with unpublished CoIP experiments we could confirm the MGME1- POLGA interaction. However, we were not able to identify additional interacting partners". I recognize that the way of describing/interpreting the data in the paragraph is unchanged after their having negative results (other than POLGA) on the interaction issue with their CoIP. The result could be a suggestion (but not evidence) of transient nature of the interaction of MGME1 and SSBP1, POLMT or Twinkle as the authors proposed, but can be also interpreted that the exogenously expressed

MGME1-BirA* biotination to SSBP1, POLMT or Twinkle was non-specific reaction. Why I insist this is because there are many black spots in Fig. 6 which, according to the authors criteria, have to be the interactors of MGME1, too. However, these proteins are not taken care of and only a small subset of them, SSBP1, POLRMT and Twinkle are conclusively claimed to interact with MEMG1 ["a number of other mitochondrial replication-related proteins that interact with MGME1 (Lines 197-198)]. Thus I feel that the way the authors describe these data in the text is not sufficiently supported by the data. As I suggested in my original comments, the authors should rewrite the paragraph carefully with consideration of how far one can say with BioID only (and with their CoIP results that no proteins excepting POLGA was pulled down via MGME1). This applies to all the sentences which is related to the data elsewhere in the manuscript (such as Lines 260-262). For example, I would suggest that Lines 197-198 should be "we identified a number of other mitochondrial replication-related proteins that could be interactors with MGME1" or "we identified a number of other mitochondrial replication-related proteins that interact with the exogenously expressed MGME1-BirA*". Or, to maintain their conclusive statement, it is necessary to provide more positive data.

Major point 10. Interpretation of S.Fig. 4

It relates to Lines 183-188, and to my original comment to S.Fig. 2 in the original manuscript (this figure is S.Fig. 4 in the new manuscript). I looked up Figure 1 of Nicholls et al HMG 2014, 23 6167-6162 and the shift of 5' ends is clear in the paper. On the other hand, data presented in S.Fig. 4 shows, in my opinion, modest changes at the 5' ends and do not appear to support the phrase "a predominance of longer 5' ends" in the absence of MGME1. It thus should be described precisely, such as "a tendency of shift towards longer 5' ends". Or, better data presentation of S.Fig. 4 may be necessary to clearly support their interpretation. In addition, it is not possible to be certain what the vertical lines in 4b stand for. Is each of them represent single clone?

Major point 11. Line 39, in relation to my major concern (5) in my original comments

"we also report a role for MGME1 in the regulation of replication and transcription termination at the end of the control region of mtDNA."

In the revised manuscript, no new data were included to support the sentence. Thus, I still maintain my opinion that it is an overstatement. They analyzed 7S DNA, transcripts in the non-coding region of mtDNA and mitochondrial transcripts. Then, from the data they speculated the possible role of MGME1, but not demonstrated it. I worry that such a conclusive wording in Abstract. I would like to suggest that "a role" should be "a possible role" and/or "report" be "propose".

Other comments

In the manuscript, the authors use the term "(linear) deletions" to the 11 kb fragments. I feel that it is confusing to readers as the word "deletion" does not usually indicate molecules but the state (of mtDNA molecule). Please be advised that the way the authors use the term "deletions" on Lines 107-110 is appropriate. Would it thus better to call the 11kb fragment as 11 kb linear sub-genomic fragment for better readability?

Line 135, Supplementary Fig. "2a" should be "2b".

Line 149-152, It would be informative for many readers if the authors add their thoughts on the specific appearance of the cox-defective cells in Mgme1-/- mice. Are they not related to premature ageing sign or something else?

Line 158, ~650 nt long is the case for human? Probably Mouse normally has shorter 7S DNA?

Line 233, "OPHOS" should be "OXPHOS".

Line 307, "I ncrease" should be "increase".

Reviewer #2 (Remarks to the Author):

The authors answered my questions satisfactorily, and I do not have any more comments. The revised manuscript has significantly improved.

Reviewer #1 (Remarks to the Author):

The revised manuscript entitled “Tissue-specific patterns of mtDNA replication stalling in mice lacking MGME1” by Matic et al. responded many of the concerns I raised. However, I still have a number of concerns on the technical aspects of data and the interpretation/description of them. They are listed below.

Major point 1. Data of 2DNAGE

If replication bubble arcs in the samples subjected to 2DNAGE are completely degraded (Fig. 9b,e and S.Fig. 7b), there would be no guarantee that replication fork arcs used for the ground of replication stalling in some, but not all, tissues of $-/-$ animals did not suffer any damage in some sample preparations but not every preparation. I respectfully disagree with the authors’ comment in their rebuttal letter that the degradation of bubble arcs does not alter their conclusion if they do not have any supporting experimental evidence of the claim. The authors also commented that replication bubble arcs are very fragile. There are, however, examples that detected bubble arcs, fork arcs and other types of replication arcs from solid tissue mtDNA preparation. For instance, Bailey et al NAR 2009:37:2327-35 showed bubble arcs from Mutator mice. Other examples are seen in Pohjoismäki et al JMB 2010:397:1144-55, Pohjoismäki et al JBC 2009:284:21446-57, Yasukawa et al EMBO J 2006:25:5358-71 and so forth. Therefore, I believe that it is much more convincing if the difference in replication is discussed with 2DNAGE that were performed with intact mtDNA preparations or at least preparations with little sign of major degradation (i.e., complete absence of replication bubbles).

Another concern on the comparison of the intensity of the fork arcs is that no information was provided on whether the same amount of mtDNA were loaded to the gels between samples. For example, in Fig. 9c, the 1N spot in $+/+$ appears to be stronger than that in $-/-$. (From Method section I am aware that 3 μ g of mtDNA was subjected to restriction enzyme digestion. However, recovery efficiency of DNA from precipitation step after the enzyme step is not always 100%). Thus, when one discusses about replication stalling based on the intensity of the arcs and spots on the arcs, a convincing way would be that the intensities of arcs/spots are standardized/normalized by those of 1N spots (I understand it is derived from non-replicating fragments) and the values of the arcs (spots)/1N are compared between $+/+$ and $-/-$. The manuscript does not describe whether the authors compared the fork arcs with such quantitative analysis. Or, the amount of mtDNA loaded to the gels should be precisely adjusted and the arcs and spots on the arcs are quantified (without 1N spot standardization, if appropriate) and compared. Also, no information is provided whether the series of 2DNAGE experiments were performed repeatedly to confirm biological reproducibility. Therefore, I would suggest the followings to confirm the technical soundness of 2DNAGE analysis.

(1) Produce higher quality 2DNAGE images with biological $n=3$ or more. Then, perform quantifications and present the results of fork arcs/1N and OL spot/1N. Or, confirm that the same amount of mtDNA preparation was run between $+/+$ and $-/-$ and quantify the intensity of the fork arcs and OL spots. For example, a same portion of fork arc between panels can be quantified for the fork arc quantification. Then present them as graphs.
(2) Or, perform above-described quantitative analysis to the existing 2DNAGE images (with biological triplicates or more).

(3) Or, for some reason if any of above quantification is not possible, at least it is necessary to present the data as Supplementary information that demonstrate fully the reproducibility of the results of Fig 9b,c,e,f and S.Fig. 7b,c (with biological triplicates or more).
Response:

We have included additional experiments in the Supplementary information (new Supplementary Fig. 9) to support the reproducibility of the 2DNAGE results. We present results from two different mice per genotype for each tissue.

Regarding loading of gels, the concentration of DNA samples was first determined and equal amounts of each DNA sample (3 ug) were restricted. Samples were then precipitated, resuspended and loaded onto the gels. This is a standard procedure for 2DNAGE, and while recovery of DNA after precipitation may not be 100 % efficient, this should not differ between samples in the same experiment. The intensity of the 1N spot (non-replicating DNA) shows the amount of loading for that panel.

We are aware that 2DNAGE technique is qualitative method and it is difficult to provide loading control for them but our conclusions concerning tissue-specific replication stalling is not based solely on the 2DNAGE but also clearly supported by an independent method, i.e. sequencing data.

Major point 2. Interpretation of brain mtDNA 2DNAGE data in the text does not appear to be supported by the data. The newly provided 2DNAGE image of brain $-/-$ sample (S.Fig 7c). has a spot on the fork arc that appears to be considerably stronger than the one on heart mtDNA (Fig.9f). The spot on S.Fig. 7c panel looks to me more similar intensity to that on Fig. 9c than that on Fig. 9f. However, this is not appreciated in the text but the gel image of brain mtDNA (S.Fig. 7c) was, like the heart case, interpreted to be consistent with the gradual decrease of read number from OH to OL. I feel that the data do not support the interpretation. To me, the brain mtDNA results suggest that the occurrence of the strong pause at OL is not related to the coverage patterns but rather suggests the presence of complex nature of replication defect variation and coverage distribution variation between tissues, which details remain unknown.

Response:

The new data from the brain mtDNA from *Mgme1*^{-/-} mice show a prominent fork arc typical of non-site-specific replication stalling. This arc is clear in both replicates, and is barely visible in control mice. Similarly, the data from heart mtDNA show a prominent fork arc in the knockout mice, which is absent from in controls. In these two tissues, an independent method (deep sequencing) shows a gradual decrease in read numbers in the direction of mtDNA replication around the major arc. In contrast, in liver mtDNA the intensity of the fork arcs is comparable between the control mice and the *Mgme1*^{-/-} mice, and consistent with this result deep sequencing shows no gradual decrease in read number between O_H and O_L. Thus, the data from two independent methods (2DNAGE and deep sequencing) are entirely consistent with the conclusions we present. We agree with the reviewer that there is a greater level of replication pausing around O_L in the brain samples in comparison with heart samples, and this finding further demonstrates the complex nature of the replication defects in different tissues.

Major point 3. Re: response to my major concern (2) in my original comment Figure presentation of photographs of the animals would be necessary to support the statement “*Mgme1* $-/-$ mice had a normal gross appearance (Line 96)”. I believe that such photos are available as some of them are nicely shown in the rebuttal letter.

Response:

We have now included pictures of the 70 weeks old control and knockout animals (Suppl. Fig. 1).

Major point 4. Identity of western blotting band. It relates to Fig. 1c, and to my original comment to Line 95 in the original manuscript. I am afraid that the new data provided in the revised Fig. 1 does not constitute the evidence that the band indicated by the authors is MGME1 protein. Independent evidence that the indicated band is MGME1 enables the authors to confirm the successful knockout of MGME1 in their $-/-$ animals at the protein level, but disappearance of the indicated band in the $-/-$ animal itself cannot be used as the evidence that the band is MGME1. That was why I suggested siRNA knockdown of MEMG1, an independent experiment to show which band is the protein, in my original comments. I believe it is an important control experiment since the slightly faster migrating band (indicated as an asterisk) also disappears in spleen, liver and kidney of $-/-$ animals and decreases in the heart. To my eyes, the faint bands in $-/-$ of liver and kidney (the new Fig. 1c) and those in $-/-$ of heart (the old Fig. 1c) migrated at slightly different positions than the asterisk band in $+/+$ counterparts. I am aware that the faster migrating band were run at the same position in $+/+$ and $-/-$ without reduction in the $-/-$ band intensity in the image in the rebuttal letter. However, images in the old and new Fig. 1c give different impression to me. What I fear is a possibility that both bands are actually MGME1 (i.e., splice variants). This point will not overturn the whole manuscript, but since MGME1 is the target protein of this work, I believe that clear identification of MGME1 band would greatly support the work.

Response:

We have performed RNAi as suggested by the reviewer. We have used three different siRNAs (targeting either exons 2, 4 or both 4 and 5) to downregulate *Mgme1* expression in mouse cells. Indeed, both bands that are recognized by the MGME1 antibody are downregulated in the RNAi samples (Figure below, panel a). The faster migrating (shorter) band shows tissue specific behaviour disappearing partly in some tissues (e.g. heart) and totally in other tissues (e.g. spleen). These two bands do not originate from differential splicing because they are also present when human MGME1-Flag is overexpressed from cDNA (no splicing possible) in HeLa cells (Figure below, panel b). Furthermore, a search of mouse gene databases gives no evidence for alternate splicing of the mouse gene encoding MGME1 (Ensembl, MGI). The lower MGME1 band seen on western blots thus represent a truncated translation or degradation product of full-length MGME1.

Major point 5. Re: response to my major concern (3) Since most of quantification data are provided after revision, the manuscript reads better now. However, the presentation is not sufficiently clear as important information for understanding the figures is missing (explained below). Fig. 2c: Does mtDNA signal (Line 647) mean “full length mtDNA signal”? Since mtDNA from $-/-$ animals show prominent sub-16kb fragments in Fig. 2b, it is informative to clearly

describe this point. Also, describe the details of the data used to produce the graph. Describe whether n=3 is biological repeats or not.

Response:

we clarified this point in our revised manuscript (see legend to Fig. 2).

S.Fig. 2a and b: The number (n) of the animals analyzed is missing. Also, describe whether n is biological repeats or not. I presume that the data from liver mtDNA of 70w *Mgme1* ^{+/+} and 70w *Mgme1* ^{-/-} in S.Fig.2a and S.Fig.2b were independently obtained from different mice as they are shown as the separate panels. State this point clearly.

Response:

We agree with the reviewer that this figure is not presented clearly. As the liver data of 70 weeks old mice were indeed the same in those two panels we have reorganized the figure now to avoid this double plotting of the same data (see Suppl. Fig. 2). Also, we have included a number of the biological replicates in the figure legend.

Fig. 3: Describe the details of whether n=3 is biological repeats or not. Also, describe whether “n=3” applies to all the panes a-d. (Does it also apply to a as well?)

Response:

We have clarified those details in the figure legend.

Fig. 4: Three triplicates (Line 661) means three sections from heart, colon and S-muscle from three different animals [= (technical triplicates) x (biological triplicates)]? Describe this point clearly. Also add the information of the week age of *Mgme1* ^{+/+}.

Response:

We have included these details in the figure legend and added the age of *Mgme1* ^{+/+} mice in figure 4.

Fig. 5c: Describe which images were quantified. Also, describe whether n=4 is biological repeats or not. Provide the quantification data that support the sentence of Lines 173-174.

Response:

We believe that the reviewer is referring to Fig 5b. The part of the image that was quantified is shown in Fig. 5a but four additional lanes (two corresponding to each genotype) were used for quantification. We have explained in the figure legend that n represents 4 biological replicates.

To support lines 173-174, we have performed quantification of wild-type levels of mtDNA and 7S DNA that are clearly showing much faster 7S DNA decay (please see figure below). In the interest of space, we choose not to include this quantification in a manuscript.

Fig. 5d: Provide a graph for the experiment [7SDNA/mtDNA ratio (and mtDNA amount, if necessary) of – and + ddC of *Mgme1*^{+/+} and *Mgme1*^{-/-}] to quantitatively demonstrate slower rate of decline of 7S DNA against mtDNA in ^{-/-}. This would be necessary to support the authors’ statement in the second paragraph of Page 7.

Response:

We respectfully disagree with the reviewer that quantification of this experiment is needed. ddc treatment in this experiment is conducted for 3 successive days and 7S DNA was shown to have very short half-life both in tissues and cultured cells (Gensler et al., NAR 2001; Clayton et al., Cell 1982), therefore the fact that under these experimental conditions (with no de-novo synthesis) we still see 7S DNA in *Mgme1*^{-/-} is already proving our point, increased stability of this DNA species in absence of MGME1.

S.Fig. 3a-c: The number (n) of the animals analyzed is missing. Also, state whether n is biological repeats or not.

Response:

We have included those details in the figure legend.

S.Fig. 6 Describe the details of whether n=3 is biological repeats or not.

Response:

We have included these details in the figure legend.

Major point 6. Newly provided long exposure images of *polgA* mut/mut samples do not clearly show the additional bands. It relates to Line 123-125, in relation to my original comment to Lines 115-116 of the original manuscript. A long exposure image is now provided in Fig. 2. However, the lower additional bands with an asterisk specific to *polgA* mut/mut is very difficult to see. At least to me, there is no band at the exact position of lower asterisk. To support the authors' statement of this sentence, presentation of stronger images on which the two bands with asterisks can be seen easily is necessary. Similarly, it is not possible to discern a band at the position of the asterisk in S.Fig. 1. In this case, it appears that better separation of the bands is required with lower percentage gel electrophoresis, or at least please show an image that one can appreciate the presence of the band.

Response:

The finding of linear deleted mtDNA molecules of a similar size and extension in mice lacking functional MGME1 and in mtDNA mutator mice expressing mutant POLGA suggests that there is a common mechanism for the formation of linear deletions. Those deleted molecules are likely caused by persistent flaps producing ligation failure (due to nuclease deficiency or exonuclease deficient POLGA). However, these molecules are not identical and are also present in different quantities in those two mouse models, as we showed by quantification in Suppl. Fig 2D. Therefore, it is not surprising that the small mtDNA fragment isn't clearly visualized by EtBr staining and labelled by radioactive probe.

Major point 7. The statement on Line 143, "Importantly, the anemia of *Mgme1*^{-/-} mice is much milder than the anemia in mtDNA mutator mice" is not supported by any data. Please show the data, or indicate a reference paper showing the data, to support this important statement. I expect that the authors can present them without a new animal experiment as the rebuttal letter says, "the mtDNA mutator mice suffer from severe anaemia with reticulocytosis and blood haemoglobin concentration as low as 50 g/l."

Response:

We have included reference to document this claim.

Major point 8. The conclusive statement, "we found increased steady-state levels of 7S DNA in *Mgme1*^{-/-} mice (Fig. 5a, b)." on Lines 160-161 is not satisfactory supported by the data provided.

The authors replied to my comment (comment to Lines 127-128 of the original manuscript),

however I am afraid that their response does not clear my concern. The authors commented the possible reason in their response, but it is not supported by any data. I suggested the use of other probes to obtain clearer images for quantification, but whether it was attempted is not known.

Since the southern blot images of Fig. 2 and S.Fig. 1b,c show weak background smear at low molecular weight region of the gels even in $-/-$ samples compared to Fig. 5a, the authors should be able to obtain the information of 7S DNA from the longer-exposure images of Fig. 2 and S.Fig. 1b,c, if not with a different hybridization probe. In any case, I consider that the accurate quantification of 7S DNA is important because better images might give a different, if not completely different, graph (and therefore interpretation) on 7S DNA steady state levels in the case of *Mgme1* knockout mice. Or, the sentence should be rewritten to explain the data carefully (as I suggested in my original comment). For example, “although the smear around 7S DNA in *Mgme1* $-/-$ mice samples prevented us from accurate quantification of 7S DNA band, southern blot analyses suggested an increase of steady-state levels of 7S DNA against full-length mtDNA in *Mgme1* $-/-$ mice.”

Response:

We made the suggested change in the manuscript.

Major point 9. The statement regarding BioID It relates to Lines190-204, and to my original comment to Lines 156-170 of the original manuscript. The authors stated in their rebuttal letter, “with unpublished CoIP experiments we could confirm the MGME1- POLGA interaction. However, we were not able to identify additional interacting partners”. I recognize that the way of describing/interpreting the data in the paragraph is unchanged after their having negative results (other than POLGA) on the interaction issue with their CoIP. The result could be a suggestion (but not evidence) of transient nature of the interaction of MGME1 and SSBP1, POLMT or Twinkle as the authors proposed, but can be also interpreted that the exogenously expressed MGME1-BirA* biotination to SSBP1, POLMT or Twinkle was non-specific reaction. Why I insist this is because there are many black spots in Fig. 6 which, according to the authors criteria, have to be the interactors of MGME1, too. However, these proteins are not taken care of and only a small

subset of them, SSBP1, POLRMT and Twinkle are conclusively claimed to interact with MEMG1 [“a number of other mitochondrial replication-related proteins that interact with MGME1 (Lines 197-198)]. Thus I feel that the way the authors describe these data in the text is not sufficiently supported by the data. As I suggested in my original comments, the authors should rewrite the paragraph carefully with consideration of how far one can say with BioID only (and with their CoIP results that no proteins excepting POLGA was pulled down via MGME1). This applies to all the sentences which is related to the data elsewhere in the manuscript (such as Lines 260-262). For example, I would suggest that Lines 197-198 should be “we identified a number of other mitochondrial replication-related proteins that could be interactors with MGME1” or “we identified a number of other mitochondrial replication-related proteins that interact with the exogenously expressed MGME1-BirA*”. Or, to maintain their conclusive statement, it is necessary to provide more positive data.

Response:

We made the suggested change in the manuscript.

Major point 10. Interpretation of S.Fig. 4 It relates to Lines183-188, and to my original comment to S.Fig. 2 in the original manuscript (this figure is S.Fig. 4 in the new manuscript). I looked up Figure 1 of Nicholls et al HMG 2014, 23 6167-6162 and the shift of 5' ends is clear in the paper. On the other hand, data

presented in S.Fig. 4 shows, in my opinion, modest changes at the 5' ends and do not appear to support the phrase “a predominance of longer 5' ends” in the absence of MGME1. It thus should be described precisely, such as “a tendency of shift towards longer 5' ends”. Or, better data presentation of S.Fig. 4 may be necessary to clearly support their interpretation. In addition, it is not possible to be certain what the vertical lines in 4b stand for. Is each of them represent single clone?

Response:

We made the suggested change in the manuscript.

Each vertical line represents single clone that was sequenced, we have included this explanation in the figure legend.

Major point 11. Line 39, in relation to my major concern (5) in my original comments “we also report a role for MGME1 in the regulation of replication and transcription termination at the end of the control region of mtDNA.” In the revised manuscript, no new data were included to support the sentence. Thus, I still maintain my opinion that it is an overstatement. They analyzed 7S DNA, transcripts in the non-coding region of mtDNA and mitochondrial transcripts. Then, from the data they speculated the possible role of MGME1, but not demonstrated it. I worry that such a conclusive wording in Abstract. I would like to suggest that “a role” should be “a possible role” and/or “report” be “propose”.

Response:

We have reworded this claim in the abstract as suggested by the reviewer.

Other comments
In the manuscript, the authors use the term “(linear) deletions” to the 11 kb fragments. I feel that it is confusing to readers as the word “deletion” does not usually indicate molecules but the state (of mtDNA molecule). Please be advised that the way the authors use the term “deletions” on Lines 107-110 is appropriate. Would it thus better to call the 11kb fragment as 11 kb linear sub-genomic fragment for better readability?

Response:

We do agree with the reviewer and have exchanged term deletions to 11kb sub-genomic fragment.

Line 135, Supplementary Fig. “2a” should be “2b”.

Response:

We have corrected this mistake.

Line 149-152, It would be informative for many readers if the authors add their thoughts on the specific appearance of the cox-defective cells in *Mgme1*^{-/-} mice. Are they not related to premature ageing sign or something else?

Response:

Mgme1^{-/-} mice have compromised mtDNA replication and as a consequence altered mtDNA expression that is at the end reflected in disturbed respiratory function. We do not believe that this is a sign of premature ageing of knockout animals.

Line 158, ~650 nt long is the case for human? Probably Mouse normally has shorter 7S DNA?

Response:

Mouse and human 7S DNA have similar size, we have modified this in the manuscript.

Line 233, “OPHOS” should be “OXPHOS”.

Response:
We have corrected this typo.

Line 307, “I ncrease” should be “increase”.

Response:
We have corrected this typo.

Reviewer #2 (Remarks to the Author):

The authors answered my questions satisfactorily, and I do not have any more comments. The revised manuscript has significantly improved.